# FREQUENCY BANDS IN RoPE:
# BASE FREQUENCY AND CONTEXT LENGTH SHAPE
# THE INTERPOLATION–EXTRAPOLATION TRADE-OFF

**Yui Oka**[1,2]**, Istumi Saito**[2]**, Kyosuke Nishida**[1]**, Kuniko Saito**[1]
[1]Human Informatics Labs., NTT, Inc.
[2]Tohoku University
yui.oka@ntt.com

## ABSTRACT

Rotary Position Embeddings (RoPE) are widely adopted in LLMs, and it is commonly believed that larger base frequencies $\theta$ yield better long-context performance. In this paper, we show that a high-norm RoPE dimension, referred to as the "frequency band," consistently emerges across multiple models, and we focus on this band to reveal the trade-offs inherent in RoPE. We find that replacing the RoPE dimensions below the frequency band with NoPE during inference has little effect on performance, indicating that these lower-frequency dimensions are only weakly utilized. We further find that the location of the frequency band depends on the RoPE base $\theta$ and the training sequence length. Moreover, the band forms early during pre-training and persists even after context extension via position interpolation. Notably, we show that setting $\theta$ to the training length shifts the band toward lower frequencies and improves extrapolation, whereas increasing $\theta$ enhances interpolation but reduces extrapolation, revealing a clear trade-off between interpolation and extrapolation. We believe this work is a step toward a sharper understanding of positional embeddings in LLMs, with falsifiable diagnostics and practical guidance for choosing $\theta$ that support scaling to longer contexts.

## 1 INTRODUCTION

Rotary Position Embedding (RoPE) (Su et al., 2023) is a widely adopted positional encoding method in Transformer-based large language models (LLMs). It can provide an awareness of relative position via two-dimensional rotations determined by a base frequency parameter, denoted as $\theta$ hereinafter. To support longer input sequences, recent work has scaled the base frequency $\theta$ well beyond its default setting of $10,000$, typically up to $500,000$ or more (Grattafiori et al., 2024; Abdin et al., 2024). This approach is motivated by the intuition that higher base frequencies alleviate sharp decay in attention scores over relative distances (Xiong et al., 2024; Rozière et al., 2024) as well by the aim of achieving extrapolation to unseen longer contexts (Vaswani et al., 2017). However, previous research shows that scaling only RoPE's $\theta$ often fails to yield robust extrapolation (Oka et al., 2025), and thus position interpolation with fine-tuning (Peng et al., 2024; Ding et al., 2024) remains necessary to recover performance in extended contexts.

Furthermore, Barbero et al. (2025) observed clear "frequency bands" in the low-frequency dimension of queries and keys, where a frequency band refers to a dimension in which high L2-norm values occur for all tokens. However, the formation of this band has not been verified. They also showed that replacing some of the low-frequency dimensions in RoPE, corresponding to the largest $\theta$, with NoPE (Kazemnejad et al., 2023) does not affect the performance of LLMs. These results suggest that such low-frequency RoPE dimensions are nearly identical to NoPE and may not represent positional information. Figure 1 illustrates a segment of the sine wave in using RoPE. As the value of $\theta_i$ increases with $\theta = 500,000$, the sine components approach zero and the cosine components approach one across most positions, effectively resulting in matrices that closely resemble the identity matrix. Such a lack of significant variation in the encoded values may underlie the phenomena discussed above.

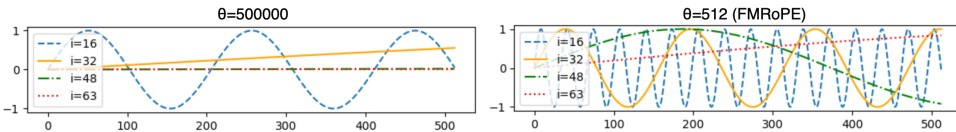

Figure 1: Sine waves of base frequencies $\theta_i$ in RoPE and a frequency-matching intervention in RoPE (FMRoPE), with training context length $L_{\text{train}} = 512$. FMRoPE sets the maximum base frequency to match the maximum sequence length in pre-training.

Theoretical reasons for $\theta$-scaling via activation decay (Xiong et al., 2024) conflict with evidence that swapping low frequencies for NoPE leaves performance unchanged (Barbero et al., 2025), revealing a deeper puzzle in RoPE's $\theta$ choice. These previous studies present a fundamental challenge to the prevailing $\theta$-scaling paradigm: **Does increasing $\theta$ truly add useful positional information or does it mainly push many RoPE dimensions into a NoPE-like form that contributes little information?** In this paper, we focus on frequency band analysis and reveal that the relationship between $\theta$ and context length from the frequency band is much closer than previously assumed.

We first present evidence that frequency bands emerge systematically across different LLMs, including Gemma (Gemma Team et al., 2024), Llama (Touvron et al., 2023; Grattafiori et al., 2024), Qwen (Yang et al., 2025), and Phi-3 (Abdin et al., 2024), and that their formation is governed by the interaction between $\theta$ and the training context length $L_{\text{train}}$. This formation is determined in the early stages of training and persists even when applying position interpolation, including YaRN (Peng et al., 2024) and LongRoPE (Ding et al., 2024)—in fact, the formation is inherited rather than corrected. Most critically, we study a frequency-matching intervention in RoPE that sets the base frequency to the training length. This shifts the frequency band toward the lowest frequencies and reveals a clear trade-off: Setting $\theta$ to the training length improves extrapolation but hurts interpolation, whereas using larger base frequencies has the opposite effect. This trade-off contradicts the prevailing notion that simply scaling $\theta$ is a universal solution for context extension.

Through extensive analysis, we provide an answer to the research question posed above: **Increasing $\theta$ does not by itself add useful positional information; rather, it mainly reallocates energy so that the dimension below the frequency band preserves positional information, while many dimensions behave similarly to NoPE and contribute little.** This improves interpolation within the training range but degrades extrapolation. Therefore, rather than treating $\theta$-scaling as universally beneficial, we emphasize the importance of considering the frequency band and the interpolation–extrapolation trade-off.

## 2 BACKGROUND

**Rotary Position Embedding (RoPE)** RoPE (Su et al., 2023) incorporates positional information directly in the self-attention mechanism by rotating the query and key vectors. The d-dimensional space is divided into $\frac{d}{2}$ subspaces, and the inner product of the rotation matrix and the query is calculated as follows.

$$\begin{bmatrix} \cos \frac{m}{\theta_i} & -\sin \frac{m}{\theta_i} \\ \sin \frac{m}{\theta_i} & \cos \frac{m}{\theta_i} \end{bmatrix} \begin{bmatrix} q_{2i-1}^m \\ q_{2i}^m \end{bmatrix}, \theta_i = \theta^{2i/d}, \tag{1}$$

where $n$ is absolute position, $q^m \in \mathbb{R}^{1 \times d}$ is the $m$-th query ($0 \le \text{m} \le L - 1$) when the number of dimensions is $d$, $i$ is the dimension ($i \in \{1, 2, \ldots, \frac{d}{2}\}$), $\theta$ is the base of RoPE, and $L$ is sequence length. The same process is also performed for the $n$-th key $k^n \in \mathbb{R}^{1 \times d}$. [1] The base $\theta$ in RoPE is relatively large and designed to represent positions exceeding the sequence length appearing during training. These positions include $\theta = 10,000$, which is based on Sinusoidal Positional Encoding (Vaswani et al., 2017) and used in the Gemma (Gemma Team et al., 2024) and Llama-2 (Touvron et al., 2023) models, $\theta = 500,000$, which is used in the Llama-3 model (Xiong et al., 2024), and $\theta = 1,000,000$, which is used in the Phi-3 model (Abdin et al., 2024).

---

[1]Note that the pretrained LLMs in Section 3 use $\theta_i = \theta^{2i/d}, i \in \{0, 1, \ldots, \frac{d}{2} - 1\}$, unlike the standard definition.

**Position Interpolation**    RoPE requires fine-tuning to handle sequences longer than the maximum sequence length $L_{\text{train}}$ appearing in pre-training. The most common approach to this fine-tuning is a position interpolation method that further expands the $\theta$ used in pre-training, and it includes YaRN (Peng et al., 2024), which determines $\theta$ with a rule-based approach, LongRoPE (Ding et al., 2024), which searches for the most suitable $\theta$ using parameter optimization, and Llama-scaling [2], which is a rule-based approach used in the Llama-3.1 model (Meta, 2024)[3].

**Frequency Bands in RoPE**    Barbero et al. (2025) revealed that there are "frequency bands" with high continuous norm values for the 2-norm $\|q^m\|_2$ and $\|k^n\|_2$ of the query and key after applying RoPE, where $q^m \in \mathbb{R}^{1 \times d}$ is the $m$-th query and $k^n \in \mathbb{R}^{1 \times d}$ is the $n$-th key when the number of dimensions is $d$. Furthermore, they also revealed that pretraining while replacing the low-frequency dimension RoPE with NoPE (Kazemnejad et al., 2023) does not change performance. This method is called p-RoPE, where $p$ is a parameter that turns the dimension into NoPE. However, their analysis focused on short texts and did not verify cases of positional interpolation or long context. Moreover, the mechanism behind the formation of the "frequency bands" remains unclear.

## 3    FREQUENCY BAND EMERGENCE IN PRETRAINED LLMS

We first investigate the frequency band identified by Barbero et al. (2025). Do similar frequency bands appear in other LLMs, or in those with base $\theta$ modified by position interpolation? To address this, we build on prior analysis (Barbero et al., 2025) and conduct further investigations across several LLMs.

### 3.1    ANALYTICAL METHODOLOGY

To measure the usage of frequencies, Barbero et al. (2025) calculated the 2-norm of key $\|k^n\|_2$. By the Cauchy-Schwarz inequality, the attention score $a_{m,n}$ between the $m$th query $q^m$ and the $n$th key $k^n$ satisfies $|\langle q^m, k^n \rangle| \leq \|q^m\|_2 \|k^n\|_2$. Therefore, to analyze the frequency components influencing the attention score, it is sufficient to examine either $\|q^m\|_2$ or $\|k^n\|_2$. We mainly examined the 2-norm of queries. Here, the 2-norm of a key is calculated as $\|k^n\|_2 = \sqrt{\sum_{j=0}^{d-1}(k_j^n)^2}$, where $d$ is the number of dimensions and $j \in \{1, 2, ..., d\}$.

**Frequency Band Index $i_{band}$**    To quantify where the frequency band appears in the key vector dimensions, we define the *band index* $i_{\text{band}}$. First, we identify the dominant frequency component at token position $n$ by selecting the dimension $i$ with the maximum 2-norm among the $\frac{d}{2}$ dimensions of the key vector $k^n$.

$$idx_n = \underset{k_i^n \in \{k_0^n, k_1^n, ..., k_{d/2-1}^n\}}{\operatorname{argmax}} (\|k_i^n\|_2) \tag{2}$$

Next, we determine the index $idx_n$ that appears most frequently in the entire sequence of length $L$. The resulting index $i\hat{d}x$ represents the dominant dimension in which the frequency bands are concentrated throughout the entire sequence.

$$i\hat{d}x = \underset{k^n \in \{k^0, k^1, ..., k^{L-1}\}}{\operatorname{argmax}} (\operatorname{count}(idx_n)) \tag{3}$$

This procedure is repeated for all heads and layers. The average of these model indices is defined as the *band index* $i_{\text{band}}$, where $0 \leq i_{\text{band}} \leq \frac{d}{2}$.

**p-RoPE**    To analyze the contribution of different frequency components in RoPE, we measured perplexity using a simplified RoPE called p-RoPE (Barbero et al., 2025), which disables low-frequency dimensions. p-RoPE applies rotation only to the top-$r$ high-frequency dimensions, interpolating between NoPE ($r = 0$) and the full RoPE ($r = 1$).

Unlike the previous studies of Barbero et al. (2025), no training was conducted in our analysis.

---

[2]transformers/modeling_rope_utils.py:L385
[3]These major position interpolations all enlarge the original $\theta$ values, as shown in Appendix I.

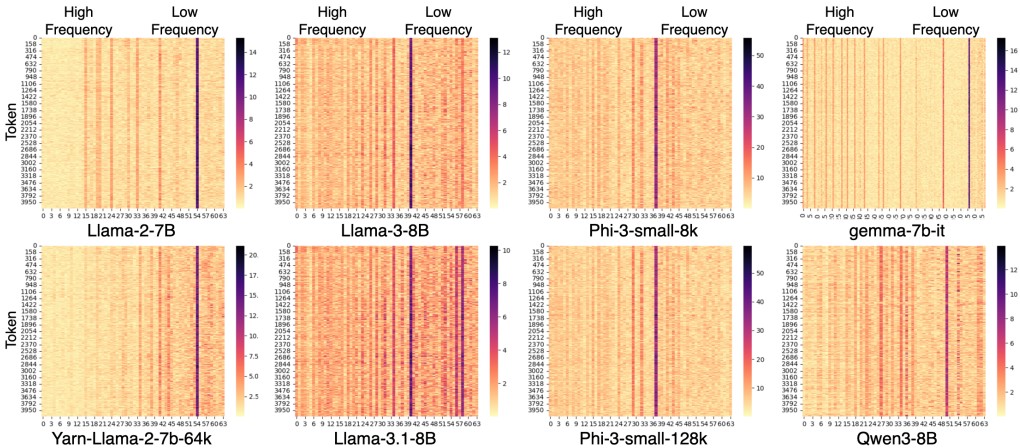

Figure 2: 2-norm plotted over 2-dimensional chunks of queries. Vertical axis represents sequence length ($L = 4096$), and horizontal axis represents each dimension index ($i \in \{0, 1, \ldots, d/2 - 1\}$) of RoPE. Note that the head dimension $d$ for the Gemma model is 256, while $d$ is 128 for other models.

Table 1: Perplexity Results with p-RoPE. 'pt' is 'Pre-train' and 'ft' is 'Fine-tuning.' YaRN, Llama3, and LongRoPE are position interpolation methods applied during fine-tuning. Note that head dimension $d$ is 256 for the Gemma model and 128 for the other models.

| Model | $L_{\text{train}}$ | | base $\theta$ | Band Index | | Perplexity with p-RoPE | | | |
|---|---|---|---|---|---|---|---|---|---|
| | pt | ft | pt | $i_{band}$ | $i_{band}/\frac{d}{2}$ | r=1.0 | r=0.9 | r=0.75 | r=0.50 |
| Gemma | 8k | - | 10000 | 116.68 | 0.91 | 2.52 | 2.70 | 81.66 | > 100 |
| Qwen3 | 40k | - | 1000000 | 51.04 | 0.79 | 6.22 | 6.22 | 6.22 | 7.46 |
| Llama-2 | 4k | - | 10000 | 53.53 | 0.84 | 2.54 | 2.58 | > 100 | > 100 |
| +YaRN | 4k | 64k | 10000 | 51.93 | 0.81 | 2.81 | 5.08 | > 100 | > 100 |
| Llama-3 | 8k | - | 500000 | 43.43 | 0.68 | 2.29 | 2.29 | 2.29 | 84.50 |
| +Llama3 | 8k | 131k | 500000 | 40.47 | 0.63 | 2.29 | 2.29 | 2.29 | 5.53 |
| Phi-3 | 8k | - | 1000000 | 36.67 | 0.57 | 2.84 | 46.11 | 46.36 | > 100 |
| +LongRoPE | 8k | 131k | 1000000 | 39.32 | 0.61 | 2.74 | 62.20 | 62.18 | > 100 |

## 3.2 EXPERIMENTAL SETTINGS

For a comprehensive analysis, we selected models that use different base models (Gemma 8B, Llama-2 7B, Llama-3 8B, Phi-3 Small, Qwen-3-8B) and different position interpolation methods (YaRN, scaling in Llama-3 model, LongRoPE). Additional details are given in Appendix A. The dataset for evaluation is the test set of Wikitext-103 (Merity et al., 2017), and the sequence length in inference is $L = 4096$ for all models.

## 3.3 RESULTS

**Do frequency bands exist in other LLMs?** Figure 2 shows the 2-norm of the queries for each model. As with Barbero et al. (2025), we extracted queries in the first layer that had semantic attention patterns in the head. First, we found that bands exist in all models, indicating that bands reflect a phenomenon that occurs generally. Next, we observed that the dimension in which the frequency band appears varies across models. Furthermore, we found that the position interpolation model inherits the bands regardless of the position interpolation method.

**Do low-frequency components of RoPE contribute to performance?** Table 1 shows frequency band index $i_{band}$ and perplexity results when varying parameter $r$ in p-RoPE across multiple language models. We also present standardized band index $i_{band}/d$ (divided by head dimension d) for unified comparison. Band index $i_{band}$ remains largely unchanged before and after position interpolation, and it aligns closely with the index of the bands shown in Figure 2, confirming consistency between our

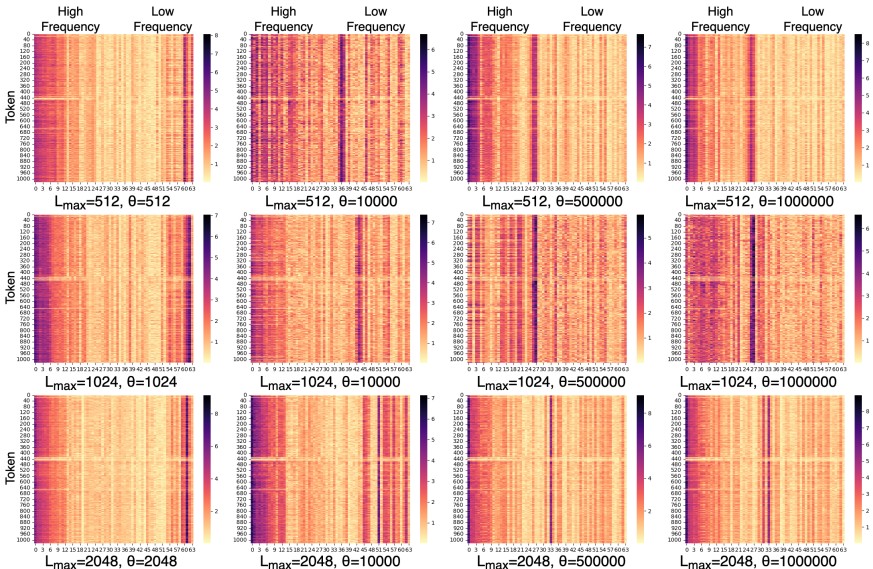

Figure 3: 2-norm plotted in the combination pattern of $(L_{\text{train}}, \theta) \in \{512, 1024, 2048\} \times \{L_{\text{train}}, 10,000; 500,000; 1,000,000\}$. Vertical axis represents sequence length ($L = 1024$), and horizontal axis represents each dimension index ($i \in \{0, 1, \ldots, d/2\}$) of RoPE.

visual and quantitative analyses. The standardized index $i_{band}/d$ decreases as $\theta$ increases, suggesting a relationship between band location and frequency determined by $\theta$. For the Gemma and Llama models, the p-RoPE results reveal that replacing RoPE in a frequency dimension lower than the band with NoPE does not degrade performance, indicating an ineffective use of low-frequency RoPE components. Conversely, Phi-3 shows performance degradation when low-frequency dimensions are replaced, regardless of band appearance, suggesting an effective use of low-frequency RoPE, possibly due to this model's block-sparse attention (Abdin et al., 2024) that alternates between dense and sparse patterns.[4]

> **Takeaways from Section 3:** In other LLMs and in models that use position interpolation, a distinct frequency band appears and remains even when the base changes. Since replacing RoPE dimensions below this frequency band with NoPE shows no measurable change, these low-frequency dimensions might not contribute to performance.

## 4 UNDERSTANDING FREQUENCY BAND FORMATION IN PRE-TRAINING

What factors cause the band index to change, and when do bands occur? To investigate the factors that determine bands, we varied RoPE's $\theta$ and max sequence length in pre-training to analyze the frequency bands via the 2-norm of the query.

### 4.1 EXPERIMENTAL SETTINGS

For pre-training, we followed the experimental settings of Press et al. (2022) and Oka et al. (2025), and we used the WikiText-103 dataset (Merity et al., 2017). A comparative evaluation was made using a Transformer-based language model (Baevski & Auli, 2019). Here, the dimensionality of the word embedding $d_{model}$ is 1024, the number of heads $N$ is 8, the dimensionality of the heads $d$ is 128, and the number of layers is 16. This implementation uses the fairseq (Ott et al., 2019)-based code. Additional details on the parameter settings are given in Appendix A. The maximum sequence length and RoPE were tested in combination with $(L_{\text{train}}, \theta) \in \{512, 1024, 2048\} \times \{L_{\text{train}}, 10,000; 500,000; 1,000,000\}$. The sequence length in inference is $L = 1024$ for all models.

---

[4]A more detailed discussion of this observation appears in Appendix D.

Table 2: Band index and perplexity with p-RoPE when sequence length in pre-training is $L = \{512, 1024, 2048\}$.

| | Base in RoPE $\theta$ | | Band Index | | Perplexity with p-RoPE | | | | |
|---|---|---|---|---|---|---|---|---|---|
| L | Train | Inference | $i_{band}$ | $i_{band}/\frac{d}{2}$ | r=1.0 | r=0.90 | r=0.75 | r=0.50 | r=0.25 |
| 512 | 512 | 512 | 60.5 | 0.94 | 19.58 | 20.18 | 24.28 | 35.11 | 98.26 |
| | 10000 | 10000 | 30.12 | 0.47 | 19.39 | 19.39 | 19.39 | 22.71 | 63.59 |
| | 500000 | 500000 | 17.00 | 0.26 | 19.35 | 19.35 | 19.35 | 19.35 | 34.46 |
| | 1000000 | 1000000 | 15.37 | 0.24 | 19.36 | 19.36 | 19.35 | 19.35 | 30.59 |
| 1024 | 1024 | 1024 | 60.25 | 0.94 | 20.07 | 20.19 | 22.37 | 32.39 | 101.97 |
| | 10000 | 10000 | 46.12 | 0.72 | 19.53 | 19.53 | 19.53 | 21.41 | 68.54 |
| | 500000 | 500000 | 18.25 | 0.28 | 19.55 | 19.55 | 19.55 | 19.55 | 34.22 |
| | 1000000 | 1000000 | 11.12 | 0.17 | 19.59 | 19.59 | 19.59 | 19.59 | 31.09 |
| 2048 | 2048 | 2048 | 60.50 | 0.94 | 21.49 | 20.99 | 21.51 | 29.56 | 94.90 |
| | 10000 | 10000 | 52.12 | 0.81 | 19.73 | 19.73 | 19.73 | 20.97 | 69.13 |
| | 500000 | 500000 | 16.62 | 0.25 | 19.71 | 19.71 | 19.71 | 19.73 | 34.98 |
| | 1000000 | 1000000 | 11.62 | 0.18 | 20.06 | 20.06 | 20.06 | 20.04 | 31.57 |

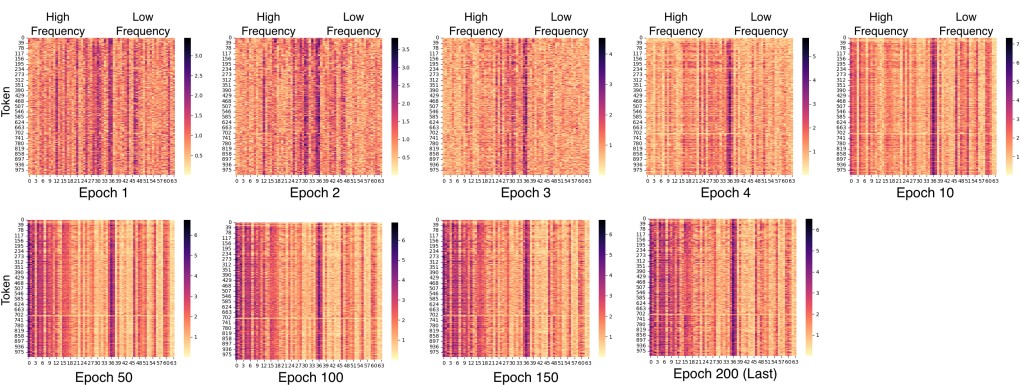

Figure 4: Plot of the 2-norm for each epoch. Vertical axis represents sequence length, and horizontal axis represents each dimension index ($i \in \{0, 1, \ldots, d/2\}$) of RoPE.

## 4.2 RESULTS

**What factors cause the band index to change?** Figure 3 shows the 2-norm map in the combination pattern. We output 2-norm maps of queries from the semantic attention head, following Section 3. First, when theta values are fixed, the index at which the band exists increases as the maximum sequence length during pre-training increases (from top to bottom of Figure 3). This suggest that the index at which the band exists depends on the maximum sequence length during pre-training. When the maximum pretraining sequence length is fixed and $\theta$ is increased ($10,000 \rightarrow 500,000 \rightarrow 1,000,000$; from left to right in Figure 3), the dominant frequency band shifts toward the lower dimensions. However, the difference between $\theta = 500,000$ and $\theta = 1,000,000$ is marginal; this similarity between the two values likely arises because both settings are already high, so further increases in $\theta$ provide little additional shift. Furthermore, when theta values were matched to the maximum sequence length during pre-training, it was found that the position of the band was near the maximum index for the head dimension.

**Band index and p-RoPE** We also investigate the band index $i_{band}$ and p-RoPE. The results when sequence length is $L_{\text{train}} = 512$ are shown in Table 2. As demonstrated in Section 3, increasing $\theta$ lowers the band index (i.e., shifts it to higher frequencies), and replacing RoPE with NoPE below this band has little impact on performance. Therefore, the frequency-band characteristics identified in Section 3 are expected to hold irrespective of model scale and training corpus.

**When do bands occur?**  We also investigated the stage when the band first appears. Figure 4 shows the key 2-norm for each epoch in the model with $L_{\text{train}}$ set to 512 and $\theta$ set to 10,000. At epoch 1, the band does not exist, and the distribution appears to be mixed with noise, but at epoch 6, the band appears from an early stage. This band is maintained until the final epoch. After the frequency band first emerges, its position remains stable throughout the remainder of training. Therefore, we can see that the band does not exist in the first stage but is still acquired by the model at an early stage during training. Epoch 6 is a stage of rapid initial convergence, during which we can see that the model acquires the band.

> **Takeaways from Section 4:** The effective dimension of RoPE is determined by the pre-training theta value and maximum sequence length, since these factors shape the band. The band emerges early in pre-training, suggesting it is a fundamental feature learned by the model.

## 5    DERIVATION OF FREQUENCY BANDS

As explained above, it has been found that the frequency band depends on the maximum sequence length and the basis. However, the mechanism itself is the core issue. This section provides a theoretical analysis to address this question. To probe the mechanism of forming the frequency band, we reduce the problem to a constrained optimization and state our guiding question: Under a fixed coefficient-norm budget, which $\theta_i$ allows the largest position-dependent variation? As a simple and informative proxy, we maximize the coordinate variance of $\cos(m\omega)$ over the window.

### 5.1    DERIVATION

**Our Goal**  We derive which RoPE pair in the query tends to concentrate energy during training, using only the maximum training sequence length $L_{\text{train}}$ and the RoPE base $\theta$. To make the argument beginner-friendly, we work with the *variance* of a single coordinate of the sinusoidal basis,

$$V(x) := \text{Var}_{m\sim\text{Unif}[0,L_{\text{train}}]}\big[\cos(m\omega)\big], \qquad x := \omega L_{\text{train}},$$

and choose the frequency that maximizes $V(x)$. Section H explains the connection to the full covariance view.

**Step 1.**  Let $m \sim \text{Unif}[0, L_{\text{train}}]$ and define $x = \omega L_{\text{train}}$. By direct integration,

$$\mathbb{E}[\cos(m\omega)] = \frac{\sin x}{x}, \qquad \mathbb{E}[\cos^2(m\omega)] = \frac{1}{2} + \frac{\sin(2x)}{4x}. \tag{4}$$

Hence, the centered variance

$$V(x) = \text{Var}[\cos(m\omega)] = \frac{1}{2} + \frac{\sin(2x)}{4x} - \left(\frac{\sin x}{x}\right)^2. \tag{5}$$

This function captures how much the $\cos$ coordinate moves across the position window. As $x \to 0$, $\cos(m\omega)$ is almost constant and $V(x) \to 0$; as $x \to \infty$, oscillations average out and $V(x) \to \frac{1}{2}$.

**Step 2.**  Differentiating Eq. (5) gives

$$V'(x) = \frac{2x^2\cos(2x) - 5x\sin(2x) + 8\sin^2 x}{4x^3}. \tag{6}$$

Stationary points satisfy $V'(x) = 0$, i.e.,

$$2x^2\cos(2x) - 5x\sin(2x) + 8\sin^2 x = 0. \tag{7}$$

Solving Eq. (7) numerically yields the smallest positive root

$$x^\star \approx 3.657210 \text{ rad} \qquad (\text{i.e., } x^\star/(2\pi) \approx 0.582 \text{ cycles}). \tag{8}$$

Here, we checked that $V(x)$ is unimodal on $(0, \infty)$ and that Eq. (8) gives the global maximum with $V(x^\star) \approx 0.54047 > \frac{1}{2}$.

**Step 3.** The continuous optimizer has angular frequency $\omega^\star = x^\star / L_{\text{train}}$. We select the RoPE pair whose grid frequency $\omega_j = \theta^{-2j/d}$ is closest to $\omega^\star$, which yields the closed-form predictor

$$j^\star \approx \frac{d}{2} \log_\theta \left( \frac{L_{\text{train}}}{x^\star} \right), \quad x^\star \approx 3.657210 . \tag{9}$$

$j^\star$ is rounded to the nearest integer; the corresponding physical dimensions are $(2j^\star, 2j^\star + 1)$.

## 5.2 DERIVED BAND LOCATION

The results of calculating $j^\star$ and $i_{band}$ in Section 3 for each model are shown in Table 3. The relationship between $j^\star$ and $i_{band}$ can be expressed as an approximately linear scaling $i_{band} \approx c \times j^\star$ with $c \approx 1.1$. This indicates that once the energy-concentrating dimension $j^\star$ is determined, the corresponding physical frequency band $i_{band}$ is essentially fixed. The small variation observed across models is likely due to differences in the query distribution rather than the model architecture. Accordingly, we proved that the position of the RoPE frequency band is predetermined by RoPE base $\theta$, training length $L_{\text{train}}$, and dimension $d$.

Table 3: Results of $j^\star$ and $i_{band}$

| Model | $j^\star$ | $i_{band}$ |
|---|---|---|
| Gemma | 107 | 116.68 |
| Llama-2 | 49 | 53.53 |
| Qwen3 | 43 | 51.04 |
| Llama-3 | 38 | 43.43 |
| Phi-3 | 36 | 36.67 |
| $\theta = L_{train}$ | 59 | - |

Furthermore, we calculated $j^\star$ when $\theta = L_{train} = 8192$ and $d = 128$. Here, $j^\star = 59$, and $c = 1.1$ yields $c \times j_{star} = 64.9$, matching the model's RoPE pair count ($\frac{d}{2} = 64$). Thus, for $\theta = L_{\text{train}}$, the band is expected to be concentrated around the 59th dimension toward the lowest-frequency dimensions.

## 5.3 CHECKING THE PREDICTED FREQUENCY-BAND POSITION

We examine whether the theorem derived in the previous section, which predicts the band location, generalizes to other choices of $\theta$ and context length. The experiments in Section 4.2 were conducted with $\theta = 512, 1024, 2048$, where the maximum training length $L$ was set equal to each value of $\theta$. We then compare the resulting empirical band positions $i_{\text{band}}$ in Section 4.2 with the theoretically predicted coordinate $j^*$. The results of calculating $j^\star$ and $i_{band}$ in Section 4.2 for each model are shown in Table 4. The relationship between $j^\star$ and $i_{band}$ can be expressed as an approximately linear scaling $i_{band} \approx c \times j^\star$ with $c \approx 1.0$.

Table 4: Results of $j^\star$ and $i_{band}$ in Section 4.2

| $\theta$ | $L$ | $j^\star$ | $i_{band}$ |
|---|---|---|---|
| 512 | 512 | 59 | 60.50 |
| 1024 | 1024 | 59 | 60.25 |
| 2048 | 2048 | 59 | 60.50 |

> **Takeaways from Section 5:** Using $x^\star \approx 3.657210$, $d$, $L_{train}$, and $\theta$, we can predict the frequency band location in advance. When $\theta = L_{train}$, the frequency band is theoretically predicted to lie at the lowest frequency dimensions.

## 6 FREQUENCY-MATCHING INTERVENTION IN RoPE

Interestingly, our analysis results suggest that higher-frequency dimensions beyond this band contribute to model performance (Section 3). However, since the frequency band is set by $\theta$ and $L_{train}$ during pretraining (Sections 4 and 5) and remains stable even with interpolation (Section 3), a natural question arises: What is the impact on model performance when the frequency band is shifted toward lower frequencies during pretraining? To explore this, we analyze a strategy we term frequency-matching intervention in RoPE (FMRoPE), where we set the base frequency parameter $\theta$ to the maximum sequence length $L_{\text{train}}$ used during pretraining. As demonstrated in Sections 4.2 and 5, this setting shifts the frequency band toward the lowest frequencies, allowing the model to leverage a broader and more effective frequency range from the start of pretraining.

Table 5: Perplexity results from Section 5. 'pt' stands for 'Pre-train' and 'ft' stands for 'Fine-tuning' in context extension with position interpolation. 'YaRN' is a position interpolation method applied during context extension. The gray area represents the FMRoPE score.

| | $L_{\text{train}}$ | | Base in RoPE $\theta$ | | Sequence Length in Inference $L$ | | | | | |
| --- | --- | --- | --- | --- | --- | --- | --- | --- | --- | --- |
| | pt | ft | Train | Inference | 512 | 1512 | 2512 | 3512 | 15512 | 25512 |
| | 512 | - | 512 | 512 | 19.58 | 21.19 | 24.20 | 27.42 | 84.75 | > 100 |
| | 512 | - | 512 | 1512 | 20.02 | **19.09** | 21.40 | 24.00 | 72.19 | >100 |
| Pre-train | 512 | - | 512 | 3512 | 21.28 | 20.27 | **20.37** | **23.00** | **66.10** | >100 |
| | 512 | - | 10000 | 10000 | 19.39 | 43.63 | 84.45 | >100 | >100 | >100 |
| | 512 | - | 500000 | 500000 | **19.35** | 40.39 | 77.90 | >100 | >100 | >100 |
| | 512 | - | 1000000 | 1000000 | **19.35** | 37.94 | 74.26 | >100 | >100 | >100 |
| | 512 | 1512 | 1512 | 1512 | 19.62 | 17.78 | **17.56** | 17.65 | 20.51 | 23.19 |
| | 512 | 1512 | 1512 | 3512 | 19.38 | 17.99 | 17.66 | **17.64** | **19.93** | 23.44 |
| Fine-tuning | 512 | 1512 | 1512 | 15512 | 21.00 | 19.74 | 19.53 | 19.48 | 20.51 | **22.41** |
| with YaRN | 512 | 1512 | 10000 | 10000 | 19.10 | 17.84 | 17.75 | 18.37 | 52.59 | 85.88 |
| | 512 | 1512 | 500000 | 500000 | 19.14 | 17.89 | 18.83 | 18.34 | 35.57 | 50.88 |
| | 512 | 1512 | 1000000 | 1000000 | **19.07** | **17.76** | 17.81 | 18.72 | 66.89 | >100 |

## 6.1 METHODOLOGY

In FMRoPE, we set the RoPE base equal to the training context length: $\theta = L_{\text{train}}$. Here, $L_{\text{train}}$ denotes the maximum sequence length used during pretraining or fine-tuning. For example, we use $\theta = 512$ during pretraining and $\theta = 1512$ during interpolation-based fine-tuning.

## 6.2 EXPERIMENTAL SETTINGS

We conducted a small-scale pre-learning and context-extension experiment, following the experimental settings of Press et al. (2022) and Oka et al. (2025) as in Section 4. The maximum sequence length during pre-training is $L_{\text{train}} = 512$, and we set $\theta = 512$. In context extension through position interpolation, we adopted YaRN (Peng et al., 2024), which is the most commonly used standard method for position interpolation. The maximum sequence length for context expansion with position interpolation is $L_{\text{train}} = 1512$. Additional details on the parameter settings can be found in Appendix A. We used perplexity as the evaluation metric. [5]

## 6.3 RESULTS

**Pre-train** We begin with the results above the dashed line in Table 5, corresponding to models without YaRN-based fine-tuning. When using conventional RoPE and FMRoPE without modification, the conventional RoPE outperforms FMRoPE. However, we observe that FMRoPE achieves better extrapolation performance. The analyses of Sections 3, 4, and 5 suggest that as more low-frequency dimensions behave like NoPE, larger $\theta$ values ($\theta \geq 10,000$) may reduce RoPE's contribution in longer contexts. In particular, the inference-time $\theta$ is adjusted to match the target sequence length (e.g., $\theta = 1512$ or $3512$), thus significantly reducing perplexity. While FMRoPE demonstrates strong extrapolation, the requirement of knowing the target sequence length at inference time poses practical limitations. Future work should explore dynamic or adaptive schemes for adjusting $\theta$ based on observed context.

**Context extension** We next examine the results below the dashed line in Table 5, corresponding to models fine-tuned with YaRN for position interpolation. FMRoPE underperforms conventional RoPE in short contexts, suggesting that FMRoPE is particularly effective in long-context or extrapolation settings but not in interpolation. FMRoPE outperforms conventional RoPE in extended sequences, achieving lower perplexity. In the FMRoPE experiment using YaRN, we found that similar trade-offs to those observed in the pre-train experiment occurred. However, as shown in Section 3, we believe this result can be expected because the frequency bands are preserved even when positional interpolation is applied.

---

[5]Comparisons with other position encodings were also conducted (Appendix F). We additionally validated our approach on a 1B-parameter model with longer contexts and evaluated downstream tasks (Appendix G).

> **Takeaways from Section 6:** Matching $\theta$ to the training length, which shift the frequency band into the lowest dimension, improves extrapolation but hurts interpolation, and this trade-off persists under position interpolation such as YaRN. Larger $\theta$ makes more low-frequency dimensions behave like NoPE, which may reduce RoPE's contribution in extrapolation.

## 7 RELATED WORK

The base $\theta$ in Sinusoidal PE (Vaswani et al., 2017) was set to $10,000$ for the purpose of enabling theoretical extrapolation. Meanwhile, Takase & Okazaki (2019) demonstrated that LRPE, which sets the base $\theta$ of SPE to the sequence length, provides robust control of output length. The $\theta$ setting adopted in this study is consistent with that setting.

RoPE's $\theta$ component has been redesigned to support context expansion with fine-tuning, including rule-based expansion of $\theta$ (Peng et al., 2024; bloc97, 2023) and learning-based or search-based frequency scaling (Chen et al., 2024; Ding et al., 2024). Furthermore, Xiong et al. (2024) reported that setting $\theta = 500,000$ during pre-training suppresses the rapid decay of attention scores between distant tokens. However, all of these methods tend to increase $\theta$, regardless of the maximum context length in pre-training. Liu et al. (2024) showed that using a smaller $\theta$ (e.g., 500) during pretraining improves extrapolation, but they did not analyze its relationship to the pretraining sequence length. In contrast, Xu et al. (2024), focusing on nearby tokens and ignoring distant context, found that such models achieve lower perplexity while still exhibiting "superficial extrapolation." Furthermore, their theoretical analysis suggests that the base frequency of RoPE governs the model's capacity to handle context length, which aligns with our findings. Barbero et al. (2025) identified RoPE frequency bands and linked them to positional heads. They also challenge the common "distance-decay" narrative and propose a modified RoPE variant.

While our visual observations overlap with Barbero et al. (2025), the core scientific questions and contributions differ substantially. We explain **where** the frequency band appears (Section 3) and **how** its position depends on the RoPE base $\theta$ and the training length $L_{train}$ (Section 4, and 5). In addition, our experiments reveal that modifying $\theta$ shifts the band toward higher or lower frequencies, and this shift leads to a clear interpolation–extrapolation trade-off (Section 6). Through theoretical analysis and controlled pre-training experiments, we identify the conditions under which bands emerge and show that their position can be predicted directly from ($\theta, L_{train}$). Our analysis also covers a broader range of models. We examine multiple pretrained LLMs (Gemma, Llama, Qwen and Phi), models after context expansion via positional interpolation (YaRN, Llama-3 scaling and LongRoPE), and models using sparse attention. In multiple pretrained LLMs analysis, we use a 4096-token context window, which is substantially longer than the 20-token window considered in Barbero et al. (2025). The results show that frequency bands persist after positional interpolation, while sparse attention changes the behavior of p-RoPE. Appendix D discuss why sparse attention is the only setting that produces a different trend. Finally, our study reveals a connection between $\theta$ and $L_{train}$. We show that modifying the value of $\theta$ induces a trade-off between interpolation and extrapolation performance.

## 8 CONCLUSION

We first showed that RoPE forms a distinct frequency band that appears across LLMs, persists after position interpolation, depends on the base $\theta$ and the training length $L_{\text{train}}$, and emerges at an early stage. Low-frequency dimensions below this band often act like NoPE and add little to performance. We derived a simple predictor by maximizing a variance proxy, yielding $x^\star \approx 3.657210$ and a grid index $j^\star$ that matches the observed band. At this point, it was theoretically understood that setting $\theta$ to $L_{\text{train}}$ would position the frequency band near the minimum frequency. Through our experiments, we found that setting $\theta$ to $L_{\text{train}}$ shifts the band to the lowest frequencies and widens the useful range, improving extrapolation while degrading interpolation. Therefore, increasing $\theta$ mostly reallocates energy rather than adding new positional information. As Practical guidance, choose $\theta \approx L_{\text{train}}$ when extrapolation is critical, and use larger $\theta$ when interpolation within the trained range is dominant. Position interpolation should be paired with a band-aware choice of $\theta$ rather than applied indiscriminately. Overall, our results connect the emergence of frequency bands to $\theta$ and $L_{\text{train}}$ and provide a new perspective for band-aware design of positional encodings in long-context LLMs.

## ACKNOWLEDGEMENTS

We sincerely thank Jun Suzuki (Tohoku University) for insightful discussions and constructive comments at the early stage of this work.

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

# A    DETAILS OF EXPERIMENTAL SETTINGS

## A.1    FREQUENCY BAND EMERGENCE IN PRETRAINED LLMs

The detailed experimental settings are described in Section 3.2. For a comprehensive analysis, we used the following models:

- `google/gemma-7b`
- `meta-Llama/Llama-2-7b`
- `NousResearch/Yarn-Llama-2-7b-64k`
- `meta-Llama/Meta-Llama-3-8B`
- `meta-Llama/Llama-3.1-8B`
- `microsoft/Phi-3-small-8k-instruct`
- `microsoft/Phi-3-small-128k-instruct`

We selected models that use different base models (Gemma, Llama, Phi-3) and different position interpolation methods (YaRN, Llama-scaling, LongRoPE). Here, the head dimension $d$ for the Gemma model is 256, and that for the other models is 128. The dataset for evaluation is the test set of Wikitext-103 (Merity et al., 2017) [6], and we used the subset of `wikitext-103-raw-v1`. This dataset is a collection of over 100 million tokens extracted from a set of articles verified as Good and Featured on Wikipedia. The subset of `wikitext-103-raw-v1` has 4358 sentences as a test set. In our analysis, we concatenated all sentences in the dataset to create a long context for measuring perplexity. The sequence length in inference is $L = 4096$ for all models.

## A.2    UNDERSTANDING FREQUENCY BAND FORMATION IN PRE-TRAINING

We described the detailed experimental settings in Section 4.1. For pre-training, we used the WikiText-103 dataset (Merity et al., 2017), which consists of over 103 million tokens of English Wikipedia articles. We performed a comparative evaluation using a Transformer-based language model (Baevski & Auli, 2019). The dimensionality of the word embedding $d_{model}$ is 1024, the number of heads $N$ is 8, the dimensionality of the heads $d$ is 128, and the number of layers is 16. This implementation used the fairseq (Ott et al., 2019)-based code provided in a previous work(Press et al., 2022), and all hyperparameters were set to the same values as those in the literature(Press et al., 2022). The number of training epochs is 205, and the batch size is 9216. The learning rate was set to 1.0, and the learning process was updated by 1e-7 every 16,000 steps. The maximum sequence length and RoPE were tested in combination with $(L_{\text{train}}, \theta) \in \{512, 1024, 2048\} \times \{L_{\text{train}}, 10,000; 500,000; 1,000,000\}$.

## A.3    FREQUENCY MATCHING IN ROTARY POSITION EMBEDDING

The detailed experimental settings are described in Section 6.2. We conducted a small-scale pre-learning and context-extension experiment. In pre-training, we used the WikiText-103 dataset (Merity et al., 2017). Furthermore, we performed a comparative evaluation using a Transformer-based language model (Baevski & Auli, 2019). Other parameter settings are the same as in Section 4.3. The maximum sequence length during pre-training is $L_{\text{train}} = 512$, and we set $\theta = 512$. In context extension achieved through position interpolation, we adopted YaRN (Peng et al., 2024), which is the most standard method for position interpolation. The maximum sequence length for context expansion with position interpolation is $L_{\text{train}} = 1512$, so we used $\theta = 1512$. Perplexity was used as the evaluation metric.

---

[6]https://huggingface.co/datasets/Salesforce/wikitext

## B LAYER-WISE VISUALIZATION OF A SINGLE ATTENTION HEAD

To verify whether the frequency-band pattern identified in Figure 2 and discussed in Section 3 is consistent across layer, we visualize the query structure for a single attention head (Head 19) across all 32 layers of Llama-3-8B. For each layer, we compute the 2-norm of the query matrix over the head dimensions and arrange the results into a two-dimensional map, following the same procedure as in Figure 2.

Figure 5 provides the full visualization. This layer-wise visualization provides a detailed view of how the characteristic frequency band emerges, shifts, or dissipates across the model. The visualizations reveal that a frequency band appears in every layer. Notably, the band becomes increasingly pronounced in deeper layers, indicating that the model progressively emphasizes a fixed set of frequencies as depth increases. In contrast, the early layers exhibit a more heterogeneous pattern, suggesting that a wider range of frequencies contributes to the representation before the model consolidates onto a narrower band. Although the exact coordinate at which the band appears varies across layers, we find that in most cases the strongest frequency concentration occurs near the dimension predicted in Section 3. Importantly, no frequency band is observed at higher frequencies beyond those identified in Section 3.

## C HEAD-WISE VISUALIZATION OF A SINGLE ATTENTION LAYER

To verify whether the frequency-band pattern identified in Figure 2 and discussed in Section 3 is consistent across head, we visualize the query structure for a single attention layer (each Layer 0 and 31) across all 32 heads of Llama-3-8B. For each head, we compute the 2-norm of the query matrix over the head dimensions and arrange the results into a two-dimensional map, following the same procedure as in Figure 2.

Figure 6 provides the full visualization in Layer 0. Figure 7 provides the full visualization in Layer 31. As reported in Section B, we observe a clear difference in the visibility of frequency bands between the shallow and deep layers. In layer 0, a well-defined band appears at almost the same coordinate across all heads. In layer 31, the band is still present, but its position shifts slightly. Even so, a faint band remains in a consistent region, which suggests that the deeper layers still inherit the influence of the original frequency band.

## D STRUCTURAL FACTORS BEHIND THE PHI-3 ANOMALY

In Section 3, only the Phi-3 model showed a tendency for p-RoPE results to differ. This section examines the reasons for this. A key difference is that Phi-3's block-sparse attention already removes a subset of long-range attention interactions by construction. This means Phi-3 relies more heavily on the positional channels that remain available. In particular, the low-frequency RoPE dimensions that encode long-range relational structure. In Llama and Gemma , multiple query heads share the same key–value projections, which creates redundancy across heads. As a result, if a subset of RoPE dimensions is replaced, other heads may still access similar positional information through shared KV projections. However, Phi-3 allocates each head a distinct block-sparse pattern. These patterns eliminate many token-pair interactions, leaving fewer pathways through which long-range positional signals can be recovered. When low-frequency RoPE dimensions are removed, Phi-3 loses one of the only mechanisms that supports long-distance reasoning under its sparsity constraints, leading to the sharp degradation observed in Table 1.

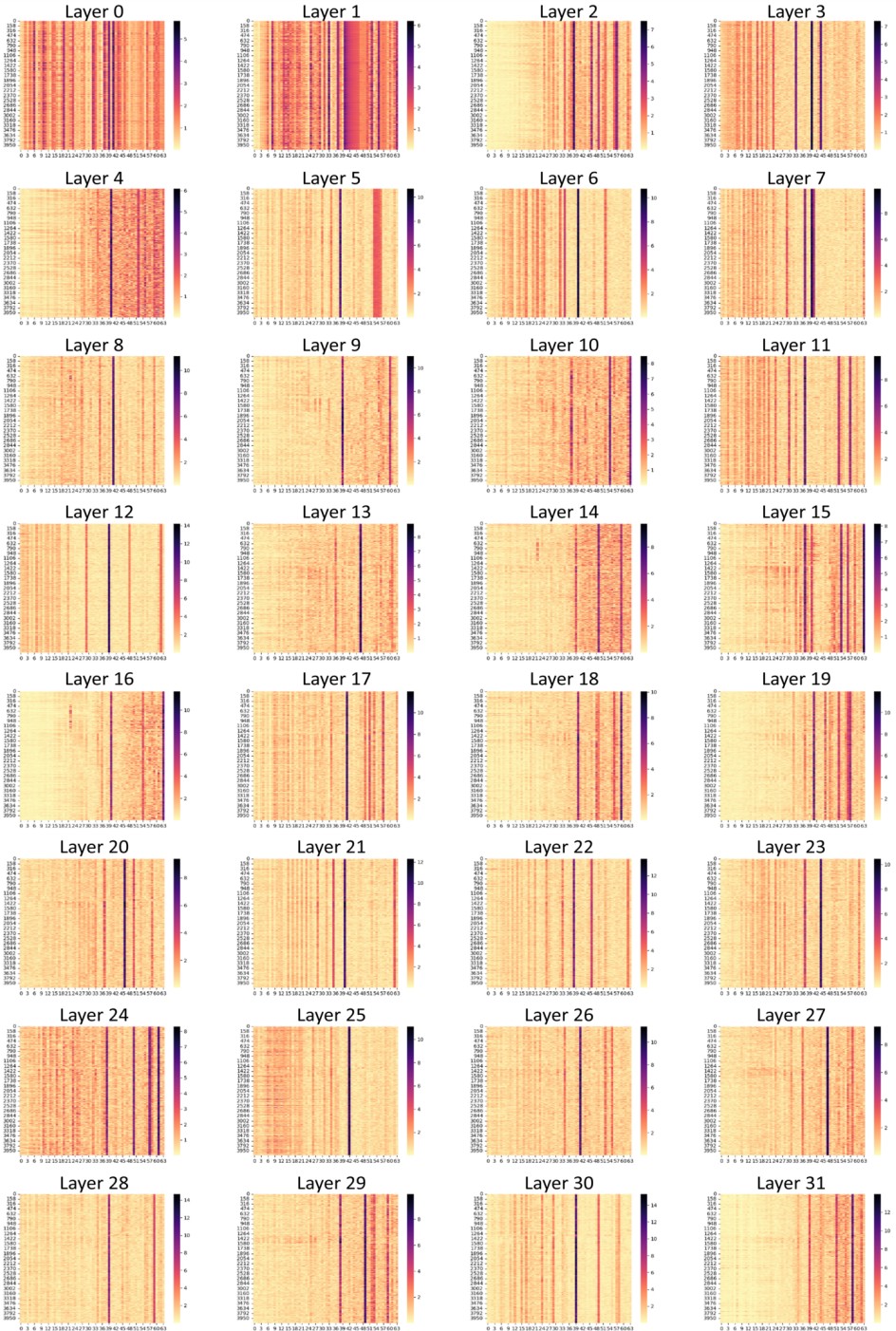

Figure 5: Layer-wise 2-norm maps of the query matrix for attention head 19 in Llama-3-8B. Each subplot shows the 2-norm plotted over 2-dimensional chunks of the query vectors, following the same visualization procedure as in Figure 2. The vertical axis corresponds to sequence length ($L = 4096$), and the horizontal axis corresponds to RoPE dimension index ($i \in 0, 1, \ldots, d/2 - 1$) with $d = 128$ for Llama-3-8B.

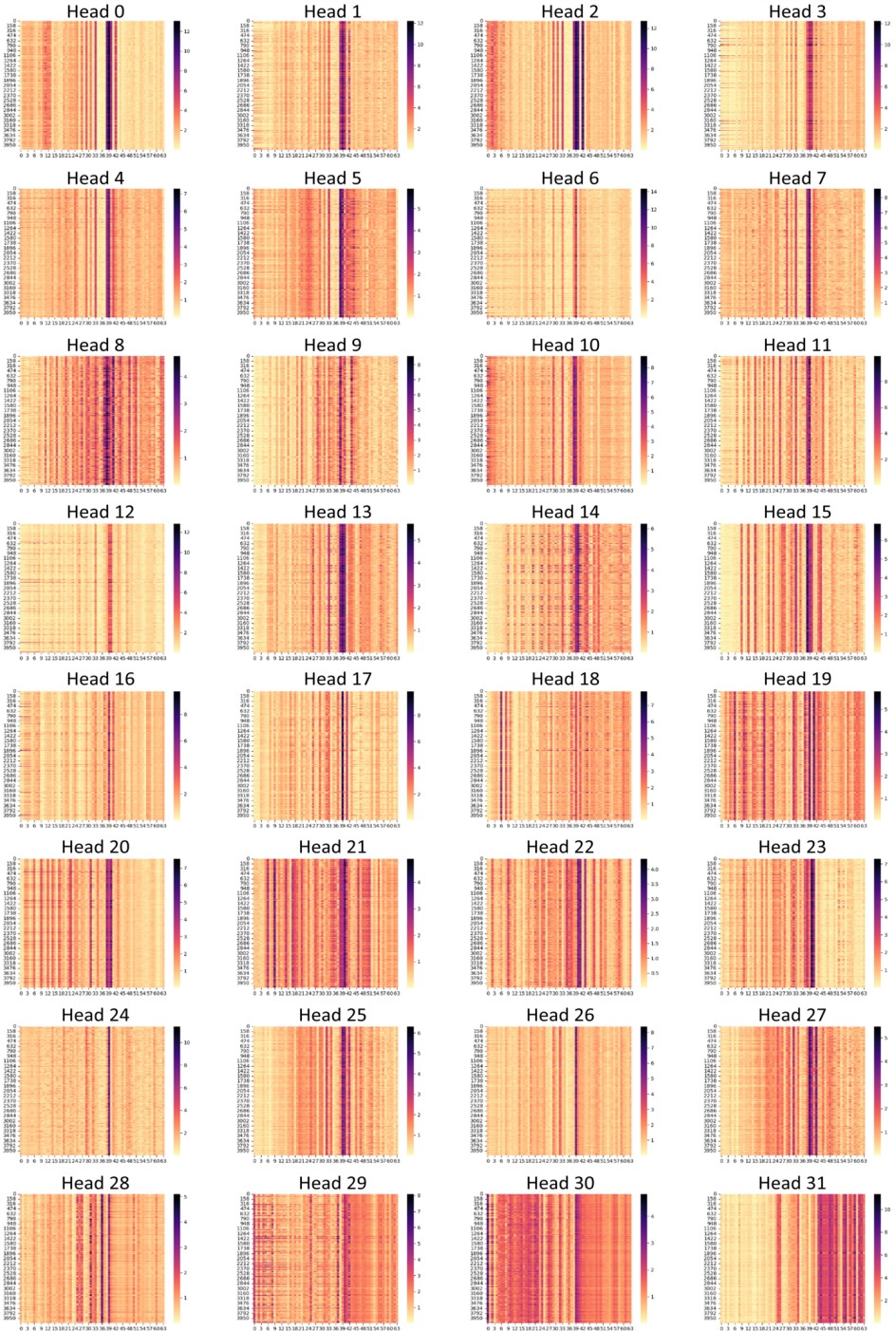

Figure 6: Head-wise 2-norm maps of the query matrix for attention layer 0 in Llama-3-8B. Each subplot shows the 2-norm plotted over 2-dimensional chunks of the query vectors, following the same visualization procedure as in Figure 2. The vertical axis corresponds to sequence length ($L = 4096$), and the horizontal axis corresponds to RoPE dimension index ($i \in 0, 1, \ldots, d/2 - 1$) with $d = 128$ for Llama-3-8B.

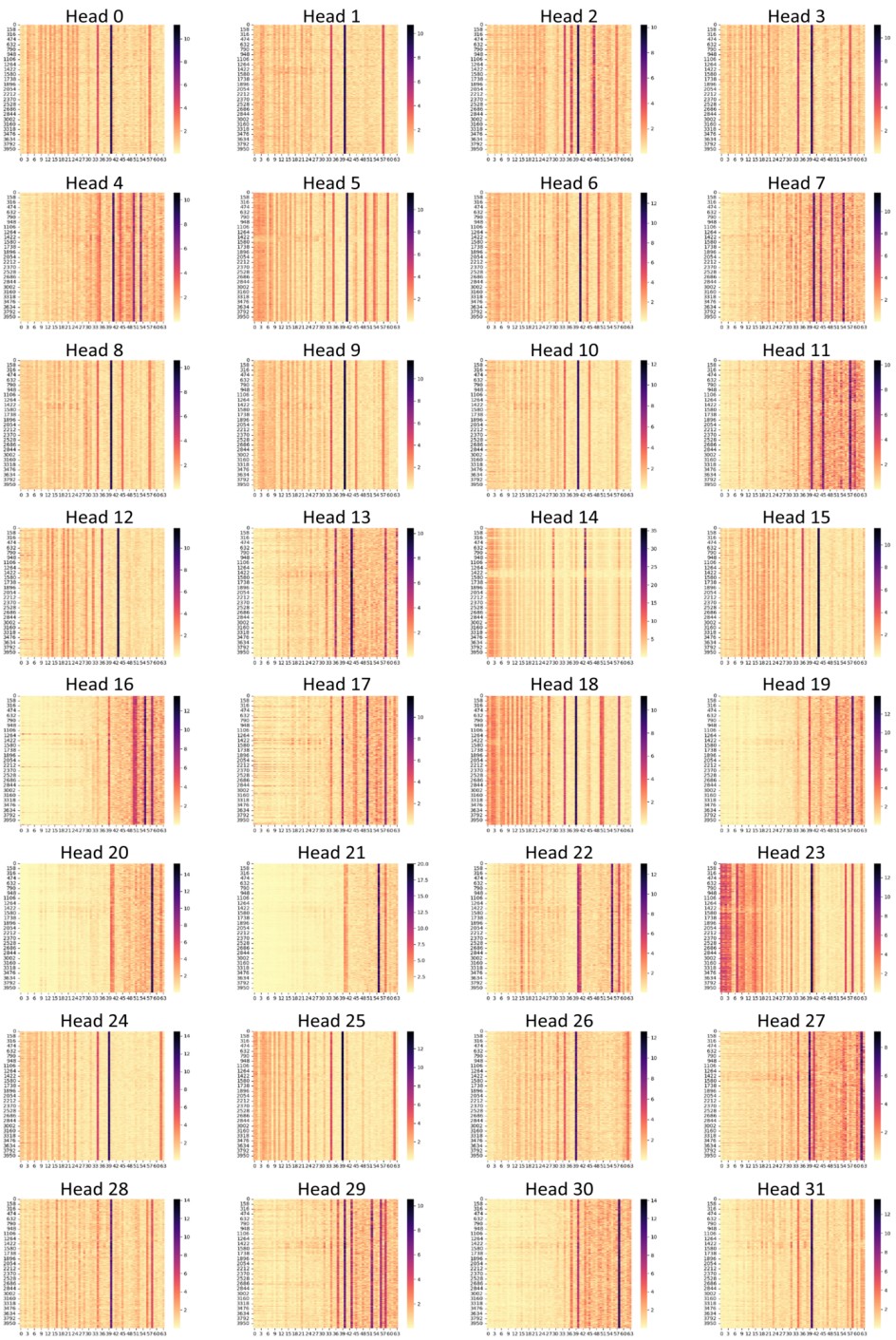

Figure 7: Head-wise 2-norm maps of the query matrix for attention layer 31 in Llama-3-8B. Each subplot shows the 2-norm plotted over 2-dimensional chunks of the query vectors, following the same visualization procedure as in Figure 2. The vertical axis corresponds to sequence length ($L = 4096$), and the horizontal axis corresponds to RoPE dimension index ($i \in 0, 1, \ldots, d/2 - 1$) with $d = 128$ for Llama-3-8B.

# E HOW POSITION HEADS BEHAVE UNDER FMRoPE

We examined the "position heads" described by (Barbero et al., 2025) under both standard RoPE and FMRoPE in Section 6. Figure 8, 9 and 10 shows 20-token attention maps for clarity. First, we confirmed that position heads exist in both RoPE and FMRoPE. Second, changing the inference-time value of $\theta$ in FMRoPE did not substantially affect these heads. Consistent with Barbero et al. (2025), these heads appear to ignore semantic content and attend purely based on relative offsets. Our results indicate that FMRoPE retains the same class of relative-position-driven heads as standard RoPE.

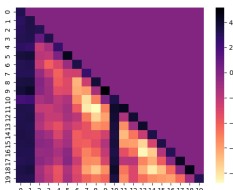

Figure 8: Position Head with RoPE (Training $\theta = 10000$, Inference $\theta = 10000$)

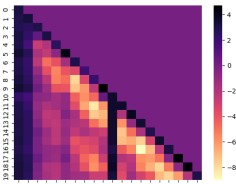

Figure 9: Position Head with RoPE (Training $\theta = 512$, Inference $\theta = 512$)

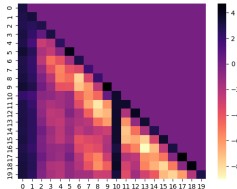

Figure 10: Position Head with RoPE (Training $\theta = 512$, Inference $\theta = 1024$)

Table 6: Perplexity results from Section 5. Here, 'pt' stands for 'Pre-train' and 'ft' stands for 'Fine-tuning' in context extension with position interpolation. 'YaRN' is a position-interpolation method applied during context extension.

| | $L_{\text{train}}$ | | base $\theta$ | | Sequence Length $L$ | | | | | |
|---|---|---|---|---|---|---|---|---|---|---|
| | pt | ft | Train | Inference | 512 | 1512 | 2512 | 3512 | 15512 | 25512 |
| NoPE | 512 | - | - | - | 21.24 | 21.32 | 46.52 | >100 | >100 | >100 |
| SPE | 512 | - | - | - | 20.02 | 77.30 | >100 | >100 | >100 | >100 |
| Transformer-XL | 512 | - | - | - | 19.98 | 18.88 | 19.02 | 19.53 | OOM | OOM |
| RPE | 512 | - | - | - | 21.20 | 21.89 | 34.77 | 74.55 | OOM | OOM |
| WaveletRPE | 512 | - | - | - | 19.20 | 17.99 | 18.00 | 18.21 | OOM | OOM |
| ALiBi | 512 | - | - | - | 19.69 | 18.53 | 18.40 | 18.43 | 18.39 | 18.39 |
| | 512 | - | 10000 | 10000 | 19.39 | 43.63 | 84.45 | >100 | >100 | >100 |
| | 512 | - | 500000 | 500000 | 19.35 | 40.39 | 77.90 | >100 | >100 | >100 |
| | 512 | - | 1000000 | 1000000 | 19.35 | 37.94 | 74.26 | >100 | >100 | >100 |
| RoPE | 512 | - | 512 | 512 | 19.58 | 21.19 | 24.20 | 27.42 | 84.75 | > 100 |
| | 512 | - | 512 | 1512 | 20.02 | 19.09 | 21.40 | 24.00 | 72.19 | >100 |
| | 512 | - | 512 | 3512 | 21.28 | 20.27 | 20.37 | 23.00 | 66.10 | >100 |
| | 512 | - | 512 | 15512 | 25.83 | 26.90 | 28.46 | 30.08 | 60.44 | 91.35 |
| | 512 | 1512 | 1512+YaRN | 1512+YaRN | 19.62 | 17.78 | 17.56 | 17.65 | 20.51 | 23.19 |
| | 512 | 1512 | 1512+YaRN | 3512+YaRN | 19.38 | 17.99 | 17.66 | 17.64 | 19.93 | 23.44 |
| | 512 | 1512 | 1512+YaRN | 15512+YaRN | 21.00 | 19.74 | 19.53 | 19.48 | 20.51 | 22.41 |
| RoPE+YaRN | 512 | 1512 | 1512+YaRN | 25512+YaRN | 21.99 | 20.89 | 20.77 | 20.84 | 21.51 | 23.19 |
| | 512 | 1512 | 10000+YaRN | 10000+YaRN | 19.10 | 17.84 | 17.75 | 18.37 | 52.59 | 85.88 |
| | 512 | 1512 | 500000+YaRN | 500000+YaRN | 19.14 | 17.89 | 18.83 | 18.34 | 35.57 | 50.88 |
| | 512 | 1512 | 1000000+YaRN | 1000000+YaRN | 19.07 | 17.76 | 17.81 | 18.72 | 66.89 | >100 |

## F COMPARISON WITH OTHER POSITION-ENCODING METHODS

### F.1 EXPERIMENTAL SETTINGS

In addition to experiment in Section 6, we also compared our method with the following position-encoding methods.

- NoPE (Kazemnejad et al., 2023)
- Sinusoidal PE (SPE) (Vaswani et al., 2017)
- Transformer-XL PE (Dai et al., 2019)
- Relative Position Representation (RPE) (Shaw et al., 2018) with clipping size 32
- Attention with Linear Biases (ALiBi) (Press et al., 2022)
- Wavelet PE (Oka et al., 2025)

For pre-training, we used the WikiText-103 dataset (Merity et al., 2017), which consists of over 103 million tokens of English Wikipedia articles. We performed a comparative evaluation using a Transformer-based language model (Baevski & Auli, 2019). The experimental setup is identical to that used in Section 6. Please refer to Appendix A.3 for details.

### F.2 PERPLEXITY RESULTS

Figure 6 presents the perplexity scores for each method. We first confirmed the effectiveness of ALiBi and WaveletPE, both of which are known for their strong extrapolation capabilities. However, methods based on relative position encoding (RPE), such as RPE itself, WaveletPE, and Transformer-XL, showed out-of-memory (OOM) errors as the sequence length increased, and these methods were unable to generate results. In contrast, ALiBi consistently maintained strong extrapolation performance even at longer sequence lengths. RoPE, on the other hand, generally exhibits lower extrapolation performance compared to other positional encoding methods. Even FMRoPE, an enhanced variant of RoPE, did not surpass the original RoPE in extrapolation ability. Nevertheless, when the context length was expanded to $L = 1512$ and the models were fine-tuned accordingly, both FMRoPE and RoPE showed improved performance relative to extrapolation-oriented PE methods. Notably, beyond $L = 1512$, FMRoPE outperformed not only RoPE but also the other PE methods.

# G   DOWNSTREAM TASK

Beyond the analyses in Section 6, we further examined FMRoPE under extended context lengths and larger model scales. In addition, we assessed performance not only in terms of perplexity but also across a suite of downstream tasks.

## G.1   EXPERIMENTAL SETUP

We trained a decoder-only Transformer with RoPE and FlashAttention. The model has $\approx 1.2$B parameters with hidden size $d_{\text{model}}{=}2048$, $n_{\text{layers}}{=}16$, $n_{\text{heads}}{=}16$, and an MLP expansion ratio of 8. We use RMSNorm without biases. Dropout is disabled throughout (`residual_dropout=0.0`, `attention_dropout=0.0`, `embedding_dropout=0.0`). The maximum training context length is 1024 tokens. Vocabulary size is 50,280 using the GPT-NeoX/OLMo Dolma v1.5 tokenizer with right-side truncation/padding; `eos_token_id= 0`, `pad_token_id= 1`. We use AdamW with $(\beta_1, \beta_2){=}(0.9, 0.95)$, $\epsilon{=}10^{-8}$, weight decay 0.1 (applied to embeddings and LayerNorm scales; `decay_norm_and_bias=true`, `decay_embeddings=true`). The peak learning rate is $6{\times}10^{-4}$ with a cosine schedule and 10,000 warmup steps; the final LR decays to $0.1\times$ the peak. We use AMP bfloat16 training with gradient clipping at 1.0. Training uses distributed data parallelism with gradient synchronization at the batch boundary. The global batch size is 512 sequences; per-device microbatch size is 4. We enable pinned memory, prefetching, and persistent dataloader workers for throughput. Checkpointing saves unsharded states every 5,000 steps; evaluation runs every 1,000 steps. We train with `flash_attention=true`. Distributed training uses `find_unused_params=false`; gradient synchronization mode is set to `batch`. We log metrics every 10 steps and monitor throughput with a moving window of 20 steps. Pretraining uses the English C4 corpus (high-quality web text) preprocessed into NumPy shards. Unless otherwise noted, we train for one epoch.

## G.2   EVALUATION METRIC

We report validation perplexity on C4 using fixed-length chunks to probe length generalization: $\{256, 512, 1024, 2048, 4096, 8192\}$ tokens. Batch size is 64. Beyond perplexity, we evaluate zero-shot performance (unless specified) on standard commonsense and QA benchmarks: PIQA (Bisk et al., 2019), HellaSwag (Zellers et al., 2019), CommonsenseQA (Talmor et al., 2019), and Social IQa (Sap et al., 2019). We additionally report Basic Arithmetic perplexity. For a more realistic long-context generation setting, we also evaluated the model on the Needle-in-a-Haystack task [7].

## G.3   RESULTS

### G.3.1   PERPLEXITY

Table 7 shows the perplexity results. When the inference length does not exceed the training length ($L \leq 1024$), all settings achieve comparable perplexity around 20. The lowest perplexity is 19.77 when training and inference both use $\theta = 10,000$. Differences appear once the inference length exceeds the pre-training context. The baseline configuration with $\theta = 1024$ shows a sharp perplexity increase to 42.36 at $L = 2048$ and diverges beyond 4096. In contrast, FMRoPE enlarges the inference base to 2048 or 8192 while keeping training at 1024, and this substantially improves extrapolation. These results show that simply enlarging the inference base frequency effectively extends the usable context without additional training. A model trained and inferred with $\theta = 10,000$ maintains competitive perplexity up to $L = 1024$ but degrades rapidly beyond that point, reaching 46.61 at $L = 2048$ and 57.83 at $L = 4096$. This observation confirms that training with an excessively high base does not guarantee long-context generalization.

### G.3.2   DOWNSTREAM TASK

Table 8 shows the downstream task results. Across all tasks, the differences among configurations are small, showing that changing the RoPE base for inference has little negative impact on general language understanding. When training and inference both use $\theta = 1024$, the model achieves strong overall accuracy with 43.96 on SocialIQA, 69.58 on PIQA, 33.66 on CommonsenseQA, 44.80 on

---

[7] https://github.com/gkamradt/LLMTest_NeedleInAHaystack

Table 7: Perplexity results from Section G. 'pt' stands for 'Pre-train' and 'ft' stands for 'Fine-tuning' in context extension with position interpolation. The gray area represents the FMRoPE score.

| | $L_{\text{train}}$ | | Base in RoPE $\theta$ | | Sequence Length in Inference $L$ | | | | | |
|---|---|---|---|---|---|---|---|---|---|---|
| | pt | ft | Train | Inference | 256 | 512 | 1024 | 2048 | 4096 | 8192 |
| Pre-train | 1024 | - | 1024 | 1024 | 23.08 | 21.02 | 19.88 | 42.36 | >100 | >100 |
| | 1024 | - | 1024 | 2048 | 23.10 | 21.05 | 19.90 | **19.33** | >100 | >100 |
| | 1024 | - | 1024 | 8192 | 23.98 | 22.07 | 21.08 | 19.85 | **19.58** | **22.86** |
| | 1024 | - | 10000 | 10000 | **23.01** | **20.94** | **19.77** | 46.61 | 57.83 | >100 |

Table 8: Downstream task results from Section G.

| Base in RoPE $\theta$ | | Downstream Task | | | | |
|---|---|---|---|---|---|---|
| Train | Inference | SocialIQA | PIQA | CommonsenseQA | HellaSwag | Arithmetic |
| 1024 | 1024 | **43.96** | 69.58 | **33.66** | 44.80 | **24.90** |
| 1024 | 2048 | 43.85 | 70.07 | **33.98** | **45.10** | 24.36 |
| 1024 | 8192 | **44.16** | 68.71 | **32.92** | 44.91 | 24.06 |
| 10000 | 10000 | 43.90 | **70.78** | 32.35 | 45.00 | 24.86 |

HellaSwag, and 24.90 on Arithmetic. Using FMRoPE with an inference base of 2048 maintains or slightly improves performance. The model reaches 70.07 on PIQA, 33.98 on CommonsenseQA, and 45.10 on HellaSwag, which are the best or nearly the best among all settings, while keeping SocialIQA and Arithmetic close to the baseline. When the inference base is further increased to 8192, performance remains stable with 44.16 on SocialIQA and 44.91 on HellaSwag, indicating that a large inference base does not harm downstream accuracy. A model trained and inferred with $\theta = 10{,}000$ achieves the highest PIQA accuracy of 70.78, although CommonsenseQA drops to 32.35.

These results show that frequency matching during inference preserves or slightly enhances downstream task performance while providing the long-context benefits demonstrated in perplexity evaluation. The findings confirm that decoupling the training and inference RoPE bases does not compromise the model's ability to perform common natural language understanding tasks.

### G.3.3 NEEDLE IN A HAYSTACK

We evaluate four settings of RoPE on the Needle-in-a-Haystack task with context lengths from 512 to 8192 tokens. The training context length is 1024 tokens. Figure 11, 12, 13 and 14 shows Needle-in-a-Haystack results. In the figures, green indicates high scores and red indicates failure.

First, the standard RoPE with base $\theta = 10000$ shows a clear boundary at the training length. Scores stay green for contexts up to 1024 tokens, which means the model interpolates well inside the training range. Beyond 1024 tokens, the scores turn red, and the model loses the ability to retrieve the needle. This behavior shows that standard RoPE does not extrapolate. Next, the model trained with base $\theta = 1024$ and inferenced with the same base shows almost the same pattern. The model keeps green scores inside 1024 tokens and turns red beyond it. However, the scores near the beginning of the sequence drop to red. When we train with base $\theta = 1024$ but inference with base $\theta = 2048$, the behavior changes. The model keeps green scores up to 2048 tokens. This result shows that the larger inference base pushes the usable range beyond the training length. A small degradation remains near the beginning, but the overall pattern suggests partial extrapolation. Finally, inference with base $\theta = 8192$ expands the effective range even further. The model keeps green scores up to 2048 tokens and shows mostly green scores around 4096 tokens. Between 4096 and 7168 tokens, some regions still appear green. These results indicate that long-range retrieval becomes possible in parts of this extended range. Overall, the model trained with =1024 exhibits clear extrapolation once the inference base is increased, demonstrating that inference-time base scaling can expand the effective context length without additional training.

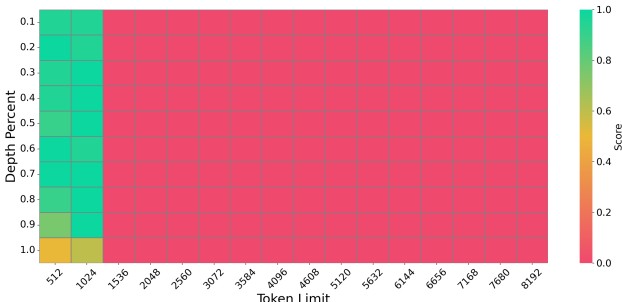

Figure 11: Needle-in-a-Haystack performance with RoPE. The maximum training context length is 1024 tokens.

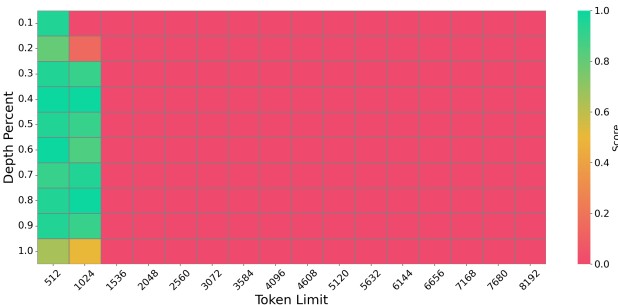

Figure 12: Needle-in-a-Haystack performance with FMRoPE ($\theta = 1024$). The maximum training context length is 1024 tokens.

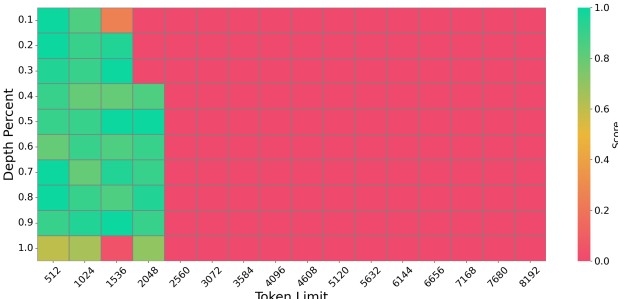

Figure 13: Needle-in-a-Haystack performance with FMRoPE ($\theta = 2048$). The maximum training context length is 1024 tokens.

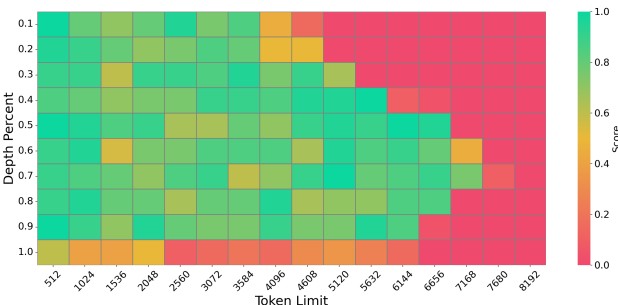

Figure 14: Needle-in-a-Haystack performance with FMRoPE ($\theta = 8192$). The maximum training context length is 1024 tokens.

### G.4    COMPUTATIONAL CONSIDERATIONS

To address computational concerns under different $\theta$ settings (RoPE vs FMRoPE), we track four metrics during training: training loss, validation loss, peak GPU memory, and throughput. As shown in Figure 15, all $\theta$ values lead to smooth and stable optimization, with no divergence in either loss curve. Peak memory usage is nearly identical across settings, and throughput varies only within normal noise levels. These results confirm that $\theta$ has no material effect on training stability, memory footprint, or efficiency. RoPE's computational cost does not depend on $\theta$. Changing $\theta$ only modifies the numerical values of the cos and sin rotations applied to each query/key pair, but the number of operations stays exactly the same. The dominant costs in inference, namely the self-attention and feed-forward matrix multiplications, remain unchanged. Therefore, inference speed and memory usage remain identical across different $\theta$ settings.

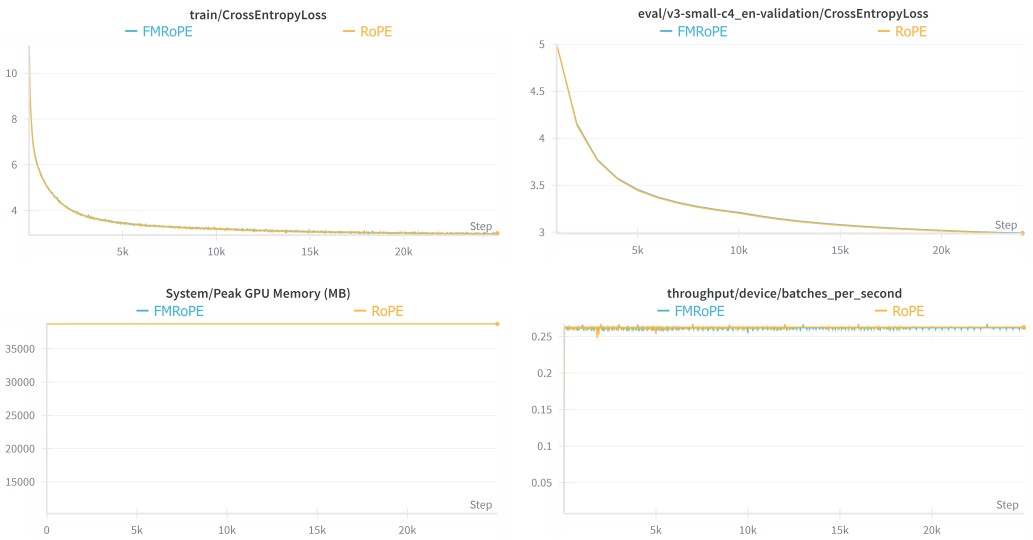

Figure 15: Training stability and computational efficiency across different base frequencies $\theta$. All plots use training steps as the x-axis. (a) Training cross-entropy loss curves. (b) Validation cross-entropy loss curves. (c) Peak GPU memory usage during training. (d) Throughput measured in batches per second.

## H   CONNECTION TO COVARIANCE VIEW (WHY THIS PROXY WORKS)

The full $2 \times 2$ covariance of the basis

$$\Sigma(\omega) \;=\; \mathrm{Cov}\left( \begin{bmatrix} \cos(m\omega) \\ \sin(m\omega) \end{bmatrix} \right) = \begin{bmatrix} \Sigma_{11} & \Sigma_{12} \\ \Sigma_{21} & \Sigma_{22} \end{bmatrix}$$

has explicit entries (with $x = \omega L_{\text{train}}$)

$$\Sigma_{11} = \tfrac{1}{2} + \frac{\sin(2x)}{4x} - \left(\frac{\sin x}{x}\right)^2, \qquad\qquad \Sigma_{22} = \tfrac{1}{2} - \frac{\sin(2x)}{4x} - \left(\frac{1-\cos x}{x}\right)^2, \qquad (10)$$

$$\Sigma_{12} = \frac{1-\cos(2x)}{4x} - \frac{\sin x}{x} \cdot \frac{1-\cos x}{x}, \qquad \Sigma_{21} = \Sigma_{12}. \qquad (11)$$

The variance we maximized is exactly the $(1,1)$ entry: $V(x) = \Sigma_{11}(x)$. If, instead, one optimizes over *all* linear combinations $A\cos(m\omega) + B\sin(m\omega)$ under a coefficient-norm budget, the centered variance is $R^2 \lambda_{\max}(\Sigma(\omega))$ by the Rayleigh–Ritz theorem. Here, $\lambda_{\max}$ represents an indicator of the maximum variance along the principal component direction of the covariance matrix and is used as a more general optimization criterion. This value can be computed via the eigenvalue decomposition of the matrix.

We additionally computed the optimal point by maximizing the largest eigenvalue $\lambda_{\max}$ of the full $2 \times 2$ RoPE matrix. This full covariance analysis yields an alternative maximizer $x \approx 4.493409$, whereas the simplified proxy gives $\tilde{x} \approx 3.657210$ in Section 4. While the numerical values differ, this discrepancy has only a minor effect on the predicted band index $i_{band}$ in Table 3 and 4, because the theoretical predictor depends on $x$ only through the logarithmic relationship in Eq (9). Substituting $x$ into this expression shifts $j$ by at most 1–2 dimensions across all values of $\theta$ used in the experiments, due to the strong dampening effect of the logarithm. As a consequence, the empirical relation between our predicted $j$ and the predicted band index $i_{band}$ (Table 3 and 4), including the near-linear fit with slope $c \approx 1.0$ - $1.1$, remains essentially unchanged.

## I   DISTRIBUTION OF $\theta_i$ IN ROPE

Figure 16 shows the distribution of $\theta_i$ when position interpolation is applied at positions 10,000, 500,000, and 1,000,000. We examined several interpolation methods, including YaRN, Llama-scaling, and LongRoPE. Overall, position interpolation tends to increase the proportion of low-frequency $\theta_i$ components.

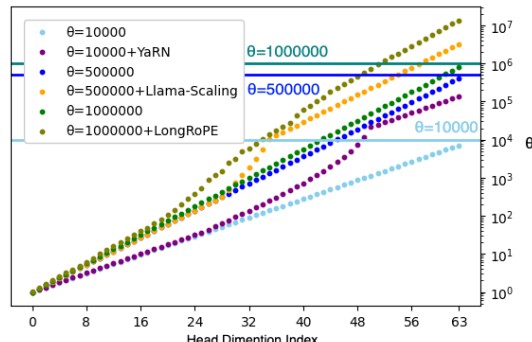

Figure 16: Distribution of $\theta_i$ values across dimensions $i$ when position interpolation is applied at positions 10,000, 500,000, and 1,000,000. The x-axis represents the dimension index $i$, and the y-axis shows the corresponding $\theta_i$ values.

## J  ANALYSIS OF LONG-TERM DECAY

To better understand interpolation and extrapolation trade-off, we next investigate the long-term decay of RoPE.

### J.1  LONG-TERM DECAY OF QUERY AND KEY

Figure 17 plots the attention logit (query–key dot product) for the first query vector in the final decoder layer across relative positions; all heads show the same trend, so we report just the first head for brevity. For large base frequencies ($\theta \geq 10,000$), the logit decays almost monotonically with distance, whereas with $\theta = 512$, no such decrease in activation is observed.

### J.2  LONG-TERM DECAY OF RoPE

To isolate the effect of $\theta$, we follow prior work (Su et al., 2023; Xiong et al., 2024) and visualize RoPE activation when both the query and key vectors are filled with ones (Figure 18, left). The original activation grows with $\theta$, confirming that larger base frequencies inject more energy into low-frequency dimensions.

Here, we hypothesize that RoPE components at frequencies higher than the band index are NoPE. To isolate the effect of the active components, we visualize the activation using only the dimensions higher than the band index in the right part of Figure 18. Surprisingly, we found that RoPE activation was reduced when theta was large. In contrast, when $\theta$ matches the sequence length, most dimensions fit within the band, resulting in relatively high activation. When the relative distance is within the maximum sequence length used during pre-training, the activation tends to be low. In contrast, for distances beyond the pre-training range, the activation becomes relatively higher. We speculate that this pattern is the reason why activation does not decrease in extrapolation in the actual activation shown in Figure 17.

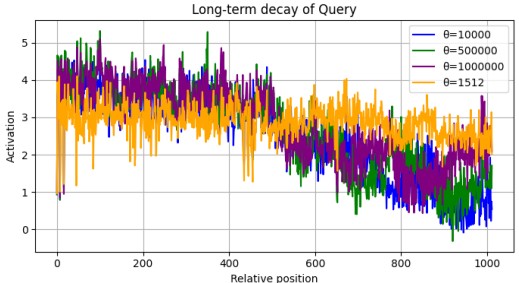

Figure 17: Attention logits (query–key dot product) for the first query vector, plotted across relative positions.

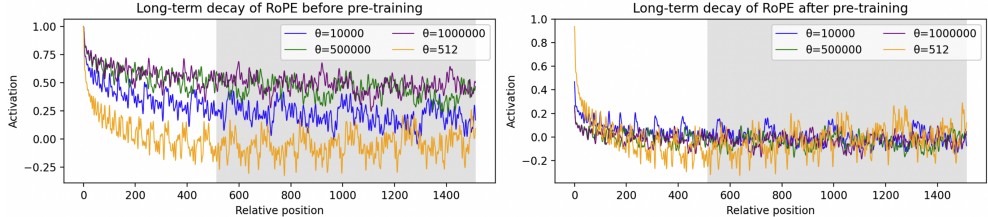

Figure 18: RoPE activation when both query and key vectors are filled with ones. Gray area indicates relative positions beyond the maximum sequence length $L_{\text{train}} = 512$ used during pre-training.

## K    LIMITATION

A potential limitation of FMRoPE is that the optimal $\theta$ may differ across pretraining, finetuning, and inference. While our analysis suggests that such cross-stage differences correspond to the same linear frequency rescaling in RoPE, and our experiments did not observe degradation from pretrain→inference mismatches (Table 5), a more systematic study at larger model scales remains an important direction for future work.

Our analysis focuses on long-context extrapolation, and we did not study multi-step reasoning tasks. Our evaluation does not fully cover larger model sizes or a wide range of long-context benchmarks. It remains an open question how FMRoPE interacts with chain-of-thought reasoning and other forms of multi-step problem solving. We leave this direction for future work.

