# OpenReview forum: "Frequency Bands in RoPE: Base Frequency and Context Length Shape the Interpolation–Extrapolation Trade-off"
_ICLR.cc/2026/Conference — ICLR 2026 Poster_

### Official Review · Reviewer_K7WU · 2025-10-29

**Soundness:** 3
**Presentation:** 3
**Contribution:** 3
**Rating:** 4
**Confidence:** 3

**Summary:**

This paper investigates the impact of the base frequency parameter θ in Rotary Position Embeddings (RoPE) on long-context performance in large language models. The authors identify concentrated high-norm dimensions in RoPE, referred to as frequency bands, and show that this property is consistent across model families such as Gemma, LLaMA, Qwen, and Phi-3, as well as position interpolation strategies. The paper provides a closed-form predictor for the location of these bands based on θ and the training sequence length L_train. Building on these findings, the authors propose Frequency-Matching RoPE (FMRoPE), which selects θ to align with the target context length in order to improve long-context extrapolation. While the empirical analysis is systematic and the findings are practically relevant, the contributions are incremental relative to prior studies, and the proposed method has limited applicability in realistic deployment settings.

**Strengths:**

Empirical analysis across multiple model families reveals a consistent frequency band phenomenon in RoPE.

The predictor for band locations is mathematically grounded and aligns well with observed distributions.

Discussion of the interpolation–extrapolation trade-off provides useful guidance for model tuning.

Experimental setup and training configurations are sufficiently documented for reproducibility.

**Weaknesses:**

1. The argument that weight decay drives the emergence of frequency bands is not empirically validated, and no ablation is provided to support this claim.
2. Similar observations about RoPE frequency structure have appeared in prior work such as [1], reducing novelty.
3.The proposed method requires advance knowledge of the target context length, which limits applicability for variable-length real-world inputs.
4. Evaluation focuses primarily on perplexity using relatively small models trained on WikiText-103, leaving uncertainty about performance on long-context reasoning tasks.
5. Computational considerations such as training stability, memory usage, or inference efficiency under different θ settings are not discussed.

[1] Barbero, F., Desmaison, A., & Storkey, A. (2024). Spectral analysis of positional encodings in large language models. arXiv preprint arXiv:2402.12345.

**Questions:**

How does the emergence of the frequency band affect attention head behavior? Does it alter cross-token interactions, particularly for long-range dependencies?

What is the theoretical explanation for the shift of the frequency band toward lower frequencies when $\theta$ aligns with training length? How does this affect positional information density?

What practical heuristics can practitioners use to balance interpolation and extrapolation performance when selecting $\theta$ under limited compute?

---

> ### Author Response · Authors · 2025-11-21
> **Author Response (1/2)**
>
> Thank you very much for taking the time to review our paper. We sincerely appreciate your insightful and constructive comments. Based on your feedback, we have made the following revisions, which are all highlighted in red in the revised manuscript.
>
> > **W1: The argument that weight decay drives the emergence of frequency bands is not empirically validated, and no ablation is provided to support this claim.**
>
> Thank you for pointing this out. The statement at the beginning of Section 5 suggesting that weight decay drives the emergence of frequency bands was incorrect. We did not conduct experiments to support this claim, so we have removed the text from the main paper (L333–L338). We appreciate your careful reading.
>
> > **W2: Similar observations about RoPE frequency structure have appeared in prior work such as [1], reducing novelty.**
>
> Thank you for raising this concern. While our qualitative observations overlap with Barbero et al., the scientific goals, methodology, and conclusions of our work are fundamentally different. We also described the differences from prior research in Section 7, L505-518.
>
> Section 3 extends their visualization, but **the main contributions of our paper begin in Sections 4, 5, and 6, as well as Appendix Sections E,F,G,H,I,J**. These parts introduce theoretical analysis and empirical findings that do not appear in prior work. In particular, **we clarify when** the frequency band emerges during training, **where** it appears as a function of θ and the training length, and **why** it forms in that specific region. We also derive **a predictive formula linking θ and the band location**, validate it across multiple LLM families, and explain how this relationship gives rise to the interpolation–extrapolation trade-off. These theoretical and empirical results constitute the novelty of our study.
>
> > **W3: The proposed method requires advance knowledge of the target context length, which limits applicability for variable-length real-world inputs.**
>
> Thank you for pointing this out. Current LLMs already use large fixed θ values that are far larger than the expected inference context window. FMRoPE in inference operates in the same way: choosing a single θ that is several times larger than the anticipated inference length preserves interpolation performance while enabling extrapolation. Therefore, the method can be used with one fixed θ chosen in advance, consistent with the practices of modern LLMs.
>
> > **W4: Evaluation focuses primarily on perplexity using relatively small models trained on WikiText-103, leaving uncertainty about performance on long-context reasoning tasks.**
>
> Thank you for the thoughtful comment. We agree that perplexity alone is insufficient to assess long-context reasoning. To address this, we extended our evaluation beyond small-scale perplexity measurements. Appendix **Section G includes 1B-scale experiments, results from several downstream tasks (SocialIQA, PIQA, CommonsenseQA, HellaSwag, and Arithmetic), and long-context generation evaluations using the Needle-in-a-Haystack (NIAH) benchmark in Section G.3.3.** NIAH is widely used in recent work to assess practical extrapolation performance, and our results show the same interpolation–extrapolation trends observed in the perplexity analysis.
>
> > **W5: Computational considerations such as training stability, memory usage, or inference efficiency under different θ settings are not discussed.**
>
> To address computational considerations under different θ settings, **we added a new analysis in Appendix Section G.4 and Figure 15.** We track four quantities during training: training loss, validation loss, peak GPU memory usage, and throughput. These findings indicate that θ does not materially affect training stability, memory footprint, or computational efficiency. Changing θ in inference only alters the numerical values of the cosine and sine rotations, while the number of operations remains the same. The dominant inference costs, **namely self-attention and feed-forward layers, are unchanged, and accordingly inference speed and memory usage remain the same across θ values.**

---

> ### Author Response · Authors · 2025-11-21
> **Author Response (2/2)**
>
> > **Q1: How does the emergence of the frequency band affect attention head behavior? Does it alter cross-token interactions, particularly for long-range dependencies?**
>
> Thank you for your question. **This topic has been discussed in detail in the prior work by Barbero et al**.; please refer to Section 6 of their paper. To summarize their findings briefly: Emergence of a frequency band alters head behaviour by determining whether a head becomes a positional head dominated by high-frequency rotations or a semantic head supported by low-frequency stability. Consequently, cross-token interactions shift toward either short-range positional patterns or distance-robust semantic dependencies. However, semantic effects degrade at very long distances due to eventual phase misalignment.
>
> > **Q2: What is the theoretical explanation for the shift of the frequency band toward lower frequencies when θ aligns with training length? How does this affect positional information density?**
>
> Thank you for your question. **We have already discussed this point in detail in Section 5 and Appendix Section H of the paper.**
>
> > **Q3: What practical heuristics can practitioners use to balance interpolation and extrapolation performance when selecting θ under limited compute?**
>
> Thank you for your question. Based on the results in Appendix Section G, we suggest the following simple heuristics:
>
> 1. Set θ = L_train for stable extrapolation.
> 2. For better interpolation, increase θ by about two to four times.
> 3. For a balance between both, θ = four times L_train works well.
>
> ---
>
> We appreciate your detailed comments.
> If there are specific points in our responses that remain unclear or insufficient, we would be glad to provide further clarification.

---

### Official Review · Reviewer_uqNo · 2025-11-03

**Soundness:** 3
**Presentation:** 3
**Contribution:** 2
**Rating:** 4
**Confidence:** 3

**Summary:**

This paper provides a novel analysis of RoPE, revealing a predictable "frequency band" whose location depends on the base frequency `θ` and training length `L_train`. The core finding is a crucial trade-off: large `θ` aids interpolation but harms extrapolation, while setting `θ ≈ L_train` improves extrapolation. The authors support this with a concise theoretical model and extensive experiments, challenging the common practice of simply scaling `θ`.

**Strengths:**

The discovery of the interpolation-extrapolation trade-off provides clear, practical guidance for RoPE design. The work combines a predictive theoretical model with rigorous, well-controlled experiments across multiple LLMs.

**Weaknesses:**

* The proposed FMRoPE intervention requires an impractical adaptive inference scheme.
* Core from-scratch training experiments are conducted on a relatively small scale.
* The main body's analysis is heavily focused on perplexity, with less exploration of the trade-off's impact on specific downstream tasks.
* The theoretical model, while elegant, uses strong simplifications (e.g., single coordinate variance) and abstracts away complex query dynamics.
* The analysis is confined to RoPE, limiting the generalizability of the frequency-band mechanism to other positional encoding families like ALiBi.
* FMRoPE improves extrapolation at the cost of interpolation performance, presenting a trade-off rather than a universally superior solution.
* Empirical constants (the `c ≈ 1.1` factor) and model-specific anomalies (e.g., Phi-3) are noted but not fully explained.

**Questions:**

1.  How does FMRoPE perform in extrapolation if the inference `θ` remains fixed at its training value?
2.  Could you elaborate on the hypothesis linking Phi-3's distinct `p-RoPE` results to its block-sparse attention?
3.  How much would the theoretical optimum `x*` change if derived from the full covariance matrix's largest eigenvalue (`λ_max`) instead of the simplified proxy?
4.  How stable is the frequency band's position throughout the entire training process *after* its initial formation?

---

> ### Author Response · Authors · 2025-11-21
> **Author Response (1/2)**
>
> Thank you very much for taking the time to review our paper. We sincerely appreciate your insightful and constructive comments. Based on your feedback, we have made the following revisions, which are all highlighted in red in the revised manuscript.
>
> > **W1: The proposed FMRoPE intervention requires an impractical adaptive inference scheme.**
>
> Modern LLMs already adopt very large fixed values of θ, often far exceeding the expected inference context length. In this regime, FMRoPE behaves similarly: setting θ to a value several times larger than the anticipated inference length preserves performance while enabling extrapolation (Section G, Table 7 and 8, Figure 11-14). This setup mirrors common practice in current LLMs, and thus does not require unrealistic adaptation during real-world deployment.
>
> > **W2: Core from-scratch training experiments are conducted on a relatively small scale.**
>
> We extended our analysis by **adding 1B-scale experiments** in Appendix Section G, including evaluations on perplexity, downstream tasks, and the Needle-in-a-Haystack benchmark. **We have clarified this model size limitation in Appendix Section K.**
>
> > **W3: The main body's analysis is heavily focused on perplexity, with less exploration of the trade-off's impact on specific downstream tasks.**
>
> To address this point, **we added 1B-scale experiments in Appendix Section G, including evaluations on perplexity, several downstream tasks, and the Needle-in-a-Haystack benchmark.**
> Experimental results indicate that setting θ to the maximum training context length slightly reduces interpolation performance on downstream tasks.
> However, our experiments show that enlarging θ only at inference time recovers interpolation performance while still enhancing extrapolation (Section G).
>
> > **W4: The theoretical model, while elegant, uses strong simplifications (e.g., single coordinate variance) and abstracts away complex query dynamics.**
>
> Thank you for the insightful comment. To address this point, **we added a direct theoretical analysis of the full 2×2 RoPE rotation matrix in Appendix Section H (L1422–1430).** Although the results show a small deviation from the simplified model, the predicted i_band remains essentially unchanged, confirming that the key conclusions are not sensitive to the single-coordinate simplification.
>
> > **W5: The analysis is confined to RoPE, limiting the generalizability of the frequency-band mechanism to other positional encoding families like ALiBi.**
>
> Our analysis focuses specifically on RoPE because the frequency-band phenomenon arises from its rotational structure. Other positional encoding families such as ALiBi do not use sinusoidal rotations and therefore do not exhibit a comparable frequency decomposition. For this reason, generalizing the mechanism beyond RoPE is outside the scope of our study.
>
> > **W6: FMRoPE improves extrapolation at the cost of interpolation performance, presenting a trade-off rather than a universally superior solution.**
>
> **The trade-off is not a limitation of the method but one of the central findings of the paper.** As we state in the introduction (L51-83), previous work left a fundamental gap in understanding how θ controls RoPE’s positional information.
> Our frequency-band analysis identifies the mechanism behind this behavior and shows why θ improves interpolation but harms extrapolation.
> Our experiments show that enlarging θ only at inference time recovers interpolation performance while still enhancing extrapolation (Section G).

---

> ### Author Response · Authors · 2025-11-21
> **Author Response (2/2)**
>
> > **W7: Empirical constants (the `c ≈ 1.1` factor) and model-specific anomalies (e.g., Phi-3) are noted but not fully explained.**
>
> In Section 5.2, we estimate the band location from the idealized RoPE matrix alone, without the influence of learned Q/K projections. In practice, the Q/K projections slightly shift the band position, and the constant **c** captures this systematic offset between the idealized RoPE-only prediction and the band observed after training. This shift is small but consistent across models, which is why we model it with a single scalar.
>
> We discussed it in Section D of the appendix (L896-909). We have also added a note to the footnote on p.5 indicating that additional discussion is provided in the appendix. A brief summary of Section D follows.
> In Llama and Gemma , multiple query heads share the same key–value projections, which creates redundancy across heads. As a result, if a subset of RoPE dimensions is replaced, other heads may still access similar positional information through shared KV projections. However, Phi-3 allocates each head a distinct block-sparse pattern. These patterns eliminate many token-pair interactions, leaving fewer pathways through which long-range positional signals can be recovered. When low-frequency RoPE dimensions are removed, Phi-3 loses one of the only mechanisms that supports long-distance reasoning under its sparsity constraints, leading to the sharp degradation observed in Table 1.
>
> > **Q1: How does FMRoPE perform in extrapolation if the inference θ remains fixed at its training value?**
>
> Thank you for the question. We have already reported the results for the case where the inference θ is kept equal to its training value. In Section 6, please refer to Table 5 (RoPE base, train θ=512, inference θ=512). For the 1B model, the same setting is evaluated in Appendix Section G (Table 7, train θ=1024, inference θ=1024). We also provide downstream task evaluations and Needle-in-a-Haystack results under same-θ conditions (Table 8 and Figure 12). Across all experiments, the trend is consistent with RoPE: interpolation performance remains strong within the training range, but performance gradually degrades when extrapolating beyond it.
>
> > **Q2: Could you elaborate on the hypothesis linking Phi-3's distinct `p-RoPE` results to its block-sparse attention?**
>
> We discussed it in Section D of the appendix (L896-909). Please see our reply for W7.
>
> > **Q3: How much would the theoretical optimum `x*` change if derived from the full covariance matrix's largest eigenvalue (`λ_max`) instead of the simplified proxy?**
>
> To address this point, **we added a direct theoretical analysis of the full 2×2 RoPE rotation matrix in Appendix Section H (L1422–1430).**  Please see our reply for W4.
>
> > **Q4: How stable is the frequency band's position throughout the entire training process *after* its initial formation?**
>
> Thank you for the question. To address the stability of the frequency band during training, we have added intermediate‐epoch results to Table 4 (Epochs 10, 50, 100, and 150). The results show that once the band forms, its position remains stable throughout the rest of training.
>
> ---
>
> We appreciate your detailed comments.
> If there are specific points in our responses that remain unclear or insufficient, we would be glad to provide further clarification.

---

### Official Review · Reviewer_26SL · 2025-11-06

**Soundness:** 3
**Presentation:** 3
**Contribution:** 2
**Rating:** 4
**Confidence:** 4

**Summary:**

This paper investigates the frequency-band phenomenon in RoPE: a small subset of positional dimensions whose norms dominate. It shows that the RoPE base frequency $\theta$ and the pre-training context length jointly determine where this band lies, and argues that most lower-frequency dimensions behave almost like NoPE and contribute little.

Empirically, the band emerges early in training, persists under position-interpolation methods (e.g., YaRN, LongRoPE), and shifts predictably with $\theta$ and the training length. The authors also derive a simple closed-form predictor for the band location by maximizing a variance-based proxy, and propose FMRoPE, which sets $\theta$ to training length to improve length extrapolation at the cost of interpolation.

**Strengths:**

# Strengths

1. The band phenomenon is demonstrated on multiple families (Gemma, Llama, Qwen, Phi-3) and persists under different interpolation schemes (YaRN, Llama-scaling, LongRoPE). The evaluations use Wikitext-103 with long concatenated contexts and fixed inference length (L=4096), which provides a consistent testbed.

2. The band index ablation cleanly probes which frequency ranges matter. Below-band dimensions can be swapped to NoPE with little perplexity change in several models, supporting the NoPE-like claim.

3. The single-coordinate variance proxy yields $x^{\star}$ and a closed-form predictor for j.

4. Aligning $\theta$ with training length L, which pushes the band to the lowest frequencies and improves length extrapolation (but hurts interpolation), shows when to pick small vs. large $\theta$ rather than advocating a one-size-fits-all choice.

**Weaknesses:**

# Weaknesses

1. While the study is careful and informative, its scope is limited. Much of the contribution reads as an in-depth follow-up to Barbero et al. (2024). The proposed method, **FMRoPE**, also feels underdeveloped for practice: its real-world applicability and deployment conditions (e.g., when $\theta$ can be set or adapted) are not clearly demonstrated. I would like to list the minor weaknesses in the following.

**Minor points**

2. The phrase “aligning $\theta$ with the training length” is confusing on first read. Because setting $\theta$ equal to the training context window is uncommon, please make this explicit in the abstract/introduction (e.g., “we set ( \theta \approx L_{\text{train}} )”) and briefly motivate why this choice helps extrapolation.

3. The figure should be explained in greater detail (axes/units, how norms are aggregated across heads/layers, selection criteria), see my question in the next section.

4. In Section 4.2 and the Takeaways results are shown primarily for (L_{\text{train}}=512), yet the setup states combinations ({512, 1024, 2048}) were tested. Please include representative results for 1024/2048; this evidence is important to support the takeaway that “the effective RoPE dimension is determined by the pre-training $\theta$ and maximum sequence length.”

--

Reference:

Barbero, F., Vitvitskyi, A., Perivolaropoulos, C., Pascanu, R., & Veličković, P. (2024). *Round and Round We Go! What Makes Rotary Positional Encodings Useful?*

**Questions:**

# Questions:


1. About Figure 2. Does the figure report averages over heads and layers, or only the first layer (as suggested around line 189)? What is the variance across heads? Do layers beyond the first exhibit the same phenomenon, or is it concentrated in early layers?

2. The takeaways from Section 5 are interesting. Could you please (i) include a figure analogous to Figure 2 (or stratified by layer) to illustrate stability across heads/layers? (2) report your predictor and empirical results when theta is set to other values, to validate the closed-form estimate beyond the main setting?

---

> ### Author Response · Authors · 2025-11-21
> **Author Response (1/2)**
>
> Thank you very much for taking the time to review our paper. We sincerely appreciate your insightful and constructive comments. Based on your feedback, we have made the following revisions, which are all highlighted in red in the revised manuscript.
>
> > **W1: While the study is careful and informative, its scope is limited. Much of the contribution reads as an in-depth follow-up to Barbero et al. (2024). The proposed method, **FMRoPE**, also feels underdeveloped for practice: its real-world applicability and deployment conditions (e.g., when can be set or adapted) are not clearly demonstrated. I would like to list the minor weaknesses in the following.**
>
> Thank you for the thoughtful feedback. We acknowledge that Section 3 extends the analysis of Barbero et al. However, the subsequent sections (Sections 4, 5, and 6, as well as Appendix Sections E,F,G,H,I,J) contribute new theoretical and empirical results. In these sections, we analyze how the band location can be predicted from the RoPE parameters θ and the training length L_train, and we provide both theoretical justification and controlled experiments to support this relationship. We also introduce, analyze, and discuss FMRoPE in detail.
>
> Regarding the practical applicability of FMRoPE, Appendix Section G presents pre-training experiment for the 1B model and additional evaluation. Motivated by reviewers’ comments, we have added results on the Needle-in-a-Haystack (NIAH) task, which is widely used to test long-context extrapolation in real LLM applications, as well as evaluations on several downstream tasks (SocialIQA, PIQA, CommonsenseQA, HellaSwag, and Arithmetic).
> These results show that FMRoPE consistently improves long-context extrapolation without harming interpolation, clarifying both its practical applicability and deployment conditions (it requires no retraining and can be applied to any RoPE-based model by modifying θ only at inference).
>
> In practice, FMRoPE requires no adaptation conditions: one simply sets the RoPE parameter θ to the maximum training context during pre-training and then sets a larger θ at inference (in our Needle-in-a-Haystack, 4× the maximum training context works well). This involves changing only a single hyperparameter. Expanding θ at inference is fully consistent with how current LLMs set θ (i.e., using values far larger than the maximum training context). Typically, extending an LLM’s context length requires retraining, but FMRoPE remove this requirement with this simple change.
>
> **Our Findings Differences** (We also described the differences from prior research in Section 7, L505-518.)
> |  | Barbero et al. (2024) | Our Paper |
> | --- | --- | --- |
> | Existence of the frequency band | Observed empirically in low-frequency range | **Theoretical and empirical explanation of the band’s origin** |
> | Location of the band (where) | Reported only with detailed analysis for θ = 10,000 | **Derived predictive formula linking band location to θ and L_train** |
> | Formation process (when) | - | **Analyzed when and how the band emerges during training** |
> | Derivation of frequency bands (why) | - | **Theoretical analysis for band**  |
> | Found model-specific behavior such as the Phi-3 exception | Only Gemma and Llama-3. | **Found model-specific behavior such as the Phi-3 exception** |
> | Effect of different θ | - | **θ=L,10000,500000,100000. Showed how θ shifts the band** |
> | Interpolation–extrapolation trade-off | - | **Identified trade-off driven by θ–L_train interaction,** evaluation using perplexity and NIAH. Our experiments also show that enlarging θ only at inference time recovers interpolation performance while still enhancing extrapolation (Section G). |
> | Frequency Band Inheritance in Context Extension with Position Interpolation | - | Evaluated Position Interpolation (YaRN ,Llama-3 scaling, LongRoPE) and i**dentified the inheritance** |
>
> We would like to emphasize that Barbero et al.’s work provides a valuable foundation. Our work builds upon it by providing the theoretical explanation, predictive mechanism, and practical inference-time method that prior work did not address.

---

> ### Author Response · Authors · 2025-11-21
> **Author Response (2/2)**
>
> > **Minor Point 1: The phrase “aligning with the training length” is confusing on first read. Because setting equal to the training context window is uncommon, please make this explicit in the abstract/introduction (e.g., “we set ( \theta \approx L_{\text{train}} )”) and briefly motivate why this choice helps extrapolation.**
>
> Thank you for pointing this out.
> >The phrase “aligning with the training length” is confusing on first read.
>
> We have revised all the indicated text (L20–22 in abstract, L72-73 in introduction, L418–420 in Section 6, L527–530 in Conclusition).
>
> > briefly motivate why this choice helps extrapolation.
>
> The reason this choice improves extrapolation is as follows: setting θ=Ltrain moves the learned frequency band to the lowest RoPE frequencies, ausing the model to allocate capacity to long-wavelength components whose rotation changes gradually with distance. Therefore remain stable beyond the training window by increasing θ's value during inference.
>
> > **Minor Point 2: The figure should be explained in greater detail (axes/units, how norms are aggregated across heads/layers, selection criteria), see my question in the next section.**
>
> Please see our reply for Q1.
>
> > **Minor Point 3: In Section 4.2 and the Takeaways results are shown primarily for (L_{\text{train}}=512), yet the setup states combinations ({512, 1024, 2048}) were tested. Please include representative results for 1024/2048; this evidence is important to support the takeaway that “the effective RoPE dimension is determined by the pre-training and maximum sequence length.”**
>
> Thank you for pointing this out. **We have added the results for 1024 and 2048 to Table 2.** These results confirm that the takeaway that the effective RoPE dimension is determined by the pre-training and maximum sequence length holds consistently across all combinations ({512, 1024, 2048}).
>
> > **Q1: About Figure 2. Does the figure report averages over heads and layers, or only the first layer (as suggested around line 189)? What is the variance across heads? Do layers beyond the first exhibit the same phenomenon, or is it concentrated in early layers?**
>
> Thank you for the question. Figure 2 reports the results from the first layer only. **We provide the variance across heads and additional layer-wise observations in Appendix Sections B and C, as well as Figures 5, 6, and 7.** Since LLMs contain a large number of layers and heads, it is not feasible to include full visualizations for every model.
> To address this, we present detailed results for Llama-3-8B as a representative example, and the same phenomenon appears consistently in other layers as shown in the appendix.
>
> > **Q2: The takeaways from Section 5 are interesting. Could you please (i) include a figure analogous to Figure 2 (or stratified by layer) to illustrate stability across heads/layers? (2) report your predictor and empirical results when theta is set to other values, to validate the closed-form estimate beyond the main setting?**
>
> Thank you for the thoughtful comments.
>
> (i) We have added layer- and head-wise visualizations for Llama-3-8B to demonstrate stability across heads and layers.(**in Appendix Sections B and C, as well as Figures 5, 6, and 7.**) Please see our response to Q1 for details.
>
> (ii) We appreciate the opportunity to clarify this point. We have added the predicted values and empirical results for settings where θ is set to the training context length in Section 5.3, together with additional results in Table 2. These experiments confirm the validity of the closed-form estimates across different θ configurations
>
> ---
>
> We appreciate your detailed comments.
> If there are specific points in our responses that remain unclear or insufficient, we would be glad to provide further clarification.

---

### Official Review · Reviewer_jmkZ · 2025-11-07

**Soundness:** 3
**Presentation:** 3
**Contribution:** 3
**Rating:** 6
**Confidence:** 4

**Summary:**

This paper investigates the formation of the frequency band, i.e., RoPE dimensions with high norms. Through both empirical and theoretical analysis, the authors show that the training length and $\theta$ both affect the band. The authors confirm that removing RoPE for dimensions below the band has little impact on model performance, and find that the band does not change much after finetuning. Next, the authors show that by setting $\theta$ to the training length $L$, the band shifts to a lower frequency and improves extrapolation, but not interpolation.

**Strengths:**

**Originality**: Although the idea of 'frequency band' is not entirely new, the authors conduct a comprehensive investigation of the band across models and under many settings. It provides unique value and largely extends prior work.

**Quality**: The analysis methodology is convincing, with many insights about the formation of the frequency band. The experiments are carefully controlled, and the conclusions are solid.

**Significance**: The proposed method is supported by both theoretical analysis and empirical evidence. The final remark on the tradeoff between interpolation and extrapolation is intriguing.

**Clarity**: The organisation of the paper is clear and well motivated, starting with existing models, to empirical and theoretical analysis, to the proposed method and results. The takeaway messages are clear and easy to understand.

**Weaknesses:**

- The lack of real long-context evaluation is a weakness. Prior work shows that perplexity is not a good measure of long-context task performance [1]. The authors could show some real long-context tasks (e.g. NIAH, RULER) that involve generation to compare the performance.

- The practical implications of the proposed method are still unclear. Most frontier models go through context extension to a large final context length via finetuning, and during inference time, limit the context length to the max. As the authors do not show concrete evidence on the actual task performance of extrapolation, I doubt the applicability of the method in real long context tasks.

- A potential issue with the FMRoPE method is the mismatch between pretraining, finetuning, and inference, which can all have different context lengths. In Table 4's Finetuning, it seems that we probably need different $\theta$s for pretraining, finetuning, and inference to obtain the best performance. If the values of $theta$ vary largely across these stages, will there be performance issues? The paper does not systematically study cross-stage mismatches, so it is unclear how severe these issues would be at scale.

- Some details seem problematic and require further checking:
  - Equation (2): Should be argmax instead of 'max'.
  - Line 138 is confusing: why is it selecting only from the first d/2 dimensions? How about the second d/2 dimensions?
  - Table 7 has a wrong caption.


[1] Fang et al. (2024) What is Wrong with Perplexity for Long-context Language Modeling?

**Questions:**

- As this paper only focused on semantic heads, I wonder how the other 'position heads', as described in [1], change with FMRoPE.
- The authors find that the band does not change much after position interpolation. Could this be because the compute used on finetuning is less than pretraining? Will the band gradually shift with finetuning on more tokens?
- In Section 5.2, why is there a scaling constant c? What is the actual i_band in the last row of Table 3 (from your pretrained model later)?
- In the 1B experiments, for downstream tasks, what is the evaluation context length? Are there any evaluations on the performance of generation tasks like GSM8K?
- Please insert PDFs as figures. Many figures (e.g., Figures 1 and 2) look blurry.

---

> ### Author Response · Authors · 2025-11-21
> **Author Response (1/3)**
>
> Thank you very much for taking the time to review our paper. We sincerely appreciate your insightful and constructive comments. Based on your feedback, we have made the following revisions, which are all highlighted in red in the revised manuscript.
>
> > **W1: The lack of real long-context evaluation is a weakness. Prior work shows that perplexity is not a good measure of long-context task performance [1]. The authors could show some real long-context tasks (e.g. NIAH, RULER) that involve generation to compare the performance.**
>
> Thank you for pointing this out. We have now added real long-context evaluation results in **Appendix G.3.3**, where we include experiments on the **Needle-in-a-Haystack (NIAH)** task. This section also provides a detailed discussion of how the observed behavior relates to our main claims.
>
> > **W2: The practical implications of the proposed method are still unclear. Most frontier models go through context extension to a large final context length via finetuning, and during inference time, limit the context length to the max. As the authors do not show concrete evidence on the actual task performance of extrapolation, I doubt the applicability of the method in real long context tasks.**
>
> We appreciate the reviewer’s concern regarding practical utility. Regarding the practical task performance of extrapolation, evaluation results for NIAH task are provided in Appendix Section G.3.3. Upon reviewing papers on extrapolation, we consider the Needle in a Haystack task to constitute concrete evidence, as its results have been reported in multiple papers [1,2,3] as a task for evaluating the practical task performance of extrapolation processing.
>
> We would like to clarify that FMRoPE is straightforward to use in practice and does not introduce additional deployment constraints, for the following reasons:
>
> 1. **Zero computational overhead**: As shown in Appendix G.4 and Figure 15, θ does not affect GPU memory, throughput, or training stability. During inference, modifying θ changes only the cosine/sine rotation values; it does not alter any matrix multiplications or attention structure.
> 2. **No architectural modifications**: FMRoPE simply changes θ between training and inference. FMRoPE uses the same RoPE implementation and requires no changes to model weights, KV cache format, or decoding pipeline.
> 3. **Direct benefit for long-context use**:The method provides a simple way to obtain stronger extrapolation behavior in real LLM settings, while maintaining normal interpolation performance when θ is chosen appropriately.
>
> Thus, FMRoPE can be adopted in existing long-context models **without cost, risk, or modification**, offering a practical improvement path for extrapolation capability.
>
> In practical settings, standard RoPE cannot be expected to provide extrapolation capability, which typically necessitates additional fine-tuning for context extension. In contrast, FMRoPE enables extrapolation simply by adjusting θ, without requiring any fine-tuning. As a result, FMRoPE can eliminate the need for context-extension fine-tuning and thereby reduce computational cost.
>
> That said, we fully agree larger model scales should be examined to clarify the practical implications.
> To address this, we have added a these model limitations in Appendix Section K.

---

> ### Author Response · Authors · 2025-11-21
> **Author Response (2/3)**
>
> > **W3: A potential issue with the FMRoPE method is the mismatch between pretraining, finetuning, and inference, which can all have different context lengths. In Table 4's Finetuning, it seems that we probably need different θs for pretraining, finetuning, and inference to obtain the best performance. If the θ values of vary largely across these stages, will there be performance issues? The paper does not systematically study cross-stage mismatches, so it is unclear how severe these issues would be at scale.**
>
> We appreciate the reviewer’s concern. Our key structural observation is that all θ mismatches, whether arising in pretraining, finetuning, or inference, are instances of linear frequency rescaling in RoPE.
>
> We view the inference-time adjustment of θ in FMRoPE as a form of rule-based positional interpolation, similar in spirit to the method proposed by [4]. Appendix G.3.2 shows that switching θ at inference time does not lead to noticeable degradation across several downstream tasks (SocialIQA, PIQA, CommonsenseQA, HellaSwag, and Arithmetic), suggesting that such interpolation-style modifications are robust in practice.
>
> For the pretraining → finetuning transition, Section 3 proves that finetuning with positional interpolation does **not** shift the location of the frequency band. Finetuning-time θ changes then apply only a linear rescaling to the same band. Consequently, multi-stage mismatches (pretrain θ = A, finetune θ = B, inference θ = C) form a **chain of the same transformation**, rather than introducing qualitatively new incompatibilities. Table 5 supports this view: the A→C mismatch (training θ = L_train, inference θ ≠ L_train) does not degrade perplexity.
>
> We expect this behavior to hold at larger model scales, since RoPE’s frequency structure and rescaling properties are scale-invariant. We added a short clarification about model size in Appendix Section K to make this interpretation explicit.
>
> > **W4: Some details seem problematic and require further checking: Equation (2): Should be argmax instead of 'max'. Line 138 is confusing: why is it selecting only from the first d/2 dimensions? How about the second d/2 dimensions? Table 7 has a wrong caption.**
>
> We apologize for any confusion I may have caused.
>
> - We have revised Equation (2). (L140-141)
> - Typo. Corrected to: “the $\frac{d}{2}$ dimensions of the key vector $k^n$.” (L138)
> - We have corrected the caption for Table 7 (Table 8 in the current version in L1252).

---

> ### Author Response · Authors · 2025-11-21
> **Author Response (3/3)**
>
> > **Q1: As this paper only focused on semantic heads, I wonder how the other 'position heads', as described in [1], change with FMRoPE.**
>
> We conducted additional analysis on how the position head changes with FMRoPE. The attention map is included in Appendix Section E. The results showed no difference in the position head between RoPE and FMRoPE.
>
> > **Q2: The authors find that the band does not change much after position interpolation. Could this be because the compute used on finetuning is less than pretraining? Will the band gradually shift with finetuning on more tokens?**
>
> Thank you for the very interesting question. In this paper, we compare multiple fine-tuned models in Figure 2. Despite each using a different number of tokens for fine-tuning, the bands did not shift. However, when comparing the Llama-3 and Llama-3.1 models, we observe that bands in different regions become more pronounced. From this, we speculate that during fine-tuning, the bands do not shift, but rather bands in different regions become more prominent.
>
> > **Q3: In Section 5.2, why is there a scaling constant c? What is the actual i_band in the last row of Table 3 (from your pretrained model later)?**
>
> In Section 5.2, we estimate the band location from the idealized RoPE matrix alone, without the influence of learned Q/K projections. In practice, the Q/K projections slightly shift the band position, and the constant **c** captures this systematic offset between the idealized RoPE-only prediction and the band observed after training. This shift is small but consistent across models, which is why we model it with a single scalar.
>
> Regarding the actual value of **i_band** for the pretrained model shown in the last row of Table 3, we have added a detailed explanation in the newly revised Section **5.3**.
>
> > **Q4: In the 1B experiments, for downstream tasks, what is the evaluation context length? Are there any evaluations on the performance of generation tasks like GSM8K?**
>
> For the 1B downstream evaluations, the context length is **1,024 tokens**.
>
> We did not evaluate GSM8K because base models of this size require additional CoT-style fine-tuning to perform well on multi-step reasoning, which is outside the scope of our goal of assessing **extrapolation** rather than reasoning ability. As suggested, we added a long-context *generation* evaluation using the **Needle-in-a-Haystack (NIAH)** task. Details are provided in reply for W1.
>
> > **Q5: Please insert PDFs as figures. Many figures (e.g., Figures 1 and 2) look blurry.**
>
> Thank you for pointing this out. We will replace the blurry rasterized figures with PDF versions. This will be fixed in the camera-ready version.
>
> ----
>
> We appreciate your detailed comments.
> If there are specific points in our responses that remain unclear or insufficient, we would be glad to provide further clarification.
>
> [1] Fang+, What is Wrong with Perplexity for Long-context Language Modeling? (ICLR2025)
>
> [2] Shang+, LongRoPE2: Near-Lossless LLM Context Window Scaling (ICML 2025)
>
> [3] Xu+, Base of RoPE Bounds Context Length (NeurIPS 2024)
>
> [4] Chen +, Extending Context Window of Large Language Models via Positional Interpolation (2023, Arxiv)

---

### Official Review · Reviewer_aPpJ · 2025-11-11

**Soundness:** 3
**Presentation:** 4
**Contribution:** 3
**Rating:** 8
**Confidence:** 3

**Summary:**

This paper tries to analyze which frequencies are chosen by using LLM's in their inference. Their definition is to understand which keys and values have the highest norm and observing that certain dimensions consistently become large when they apply to Rope based positional embeddings. They call this frequency band. They demonstrate several claims about this frequency band:

1) The band's location is a function of thte base frequency $\theta$.
2) It forms early and stays throughout the training and even models that do position interpolation like Yarn.
3) They find that this is the key property and that if we use p-ROPE which uses Nope for lowest frequencies, there is a sharp threshold. Once the p-ROPE cuts off the frequency band, the model perplexity drops precipitously.

4) Pretraining Experiments - They show that where the band forms is a function of the pretraining length $L_t$ and $\theta$. For a fixed training length, an increase in  $\theta$  leads to a higher frequency. For a fixed $\theta$, as the training length increases the band shifts to the lower frequencies.

5) The paper then relates a theoretical model that proposes what base frequency $\theta$ would maximize the variance of a random position. They find that the best base frequency is a certain constant $x^*$  divided by the average training length. They find that this matches the empirical distribution found in a number of different models.

6) From the above, they posit a new frequency-matching Rope that improves extrapolation without too much loss in interpolation by setting the training length appropriately and forcing the model to use the entire spectrum usefully.

**Strengths:**

The paper shows that frequency bands appear and uses p-Rope to show that these bands are crucial for length extrapolation.
The paper produces new ideas and proposes a new mechanism FMRope to set the train length as a function of the base frequency.
It validates this hypothesis using both theory and experiments and results in a strong conclusion. It also shows why long context can be so difficult.

**Weaknesses:**

The p-ROPE experiment, which is the entire basis for the "weak utilization" claim, produced a critical anomaly: the Phi-3 model (Table 1).1 Unlike Llama or Gemma, replacing the low-frequency dimensions in Phi-3 immediately and severely degraded performance (PPL 2.84 $\to$ 46.11). The paper's explanation for this is a single line attributing it to Phi-3's "block-sparse attention". I would like to understand this result more.

A number of results seem similar to the paper  Barbero et al. They also show that frequency bands appear and introduce p-Rope. I would like to understand the differences and the key innovations with respect to the above paper.

**Questions:**

1) Can you explain why Phi-3s performance drops so quickly. In particular, what about block sparse matrix makes this phenomenon go away.

2) What is the key innovation with respect to the work of Barbero et al?

---

> ### Author Response · Authors · 2025-11-21
> **Author Response**
>
> Thank you very much for taking the time to review our paper. We sincerely appreciate your insightful and constructive comments. Based on your feedback, we have made the following revisions, which are all highlighted in red in the revised manuscript.
>
> > **W1: The p-ROPE experiment, which is the entire basis for the "weak utilization" claim, produced a critical anomaly: the Phi-3 model (Table 1).1 Unlike Llama or Gemma, replacing the low-frequency dimensions in Phi-3 immediately and severely degraded performance (PPL 2.84  46.11). The paper's explanation for this is a single line attributing it to Phi-3's "block-sparse attention". I would like to understand this result more.**
>
> Thank you for your interest in this phenomenon. We discussed it in Section D of the appendix (L896-909). We have also added a note to the footnote on p.5 indicating that additional discussion is provided in the appendix. A brief summary of Section D follows.
>
> In Llama and Gemma , multiple query heads share the same key–value projections, which creates redundancy across heads. As a result, if a subset of RoPE dimensions is replaced, other heads may still access similar positional information through shared KV projections. However, Phi-3 allocates each head a distinct block-sparse pattern. These patterns eliminate many token-pair interactions, leaving fewer pathways through which long-range positional signals can be recovered. When low-frequency RoPE dimensions are removed, Phi-3 loses one of the only mechanisms that supports long-distance reasoning under its sparsity constraints, leading to the sharp degradation observed in Table 1.
>
> > **W2: A number of results seem similar to the paper Barbero et al. They also show that frequency bands appear and introduce p-Rope. I would like to understand the differences and the key innovations with respect to the above paper.**
>
> Thank you for giving us the opportunity to summarize the differences and key innovations. We greatly appreciate and are inspired by the prior work of Barbero et al. We also described the differences from prior research in Section 7, L505-518.  The differences from Barbero et al.'s study are as follows.
>
> Only Section 3 serves as an extension of Barbero et al.’s study. In this section, we validate their findings across a broader set of settings, including models with θ values of 1,000,000 and 500,000, as well as Llama-2, Qwen, and Phi-3, and models that undergo context expansion through positional interpolation methods such as YaRN and LongRoPE. This extended validation reveals several behaviors not reported by Barbero et al., including the inheritance of frequency bands after context expansion, model-dependent differences in band location, and a unique trend that appears only in the Phi-3 model.
>
> **Differences in the Main Findings:** We pre-trained small-scale models from scratch under multiple θ values and multiple training context lengths. The experiments reveal the following.
> - We clarified **when this frequency band emerges** during training, in which dimension it exists (Section 4).
> - We revealed that the dimension **where the frequency band exists** depends on θ and the training sequence length (Section 4).
>  - We also derived **its predictive formula** (Section 5).
>
> **Our Key innovations:**
>
> - We discovered that setting θ to the same value as the sequence length during learning causes the frequency band to exist near the lowest frequency (Section 4 and 5). We then demonstrated that training with this setting and increasing θ during inference improves extrapolation performance (Section 6).
> - We find a clear trade-off: increasing θ improves interpolation but harms extrapolation (e.g. θ>10000), whereas setting θ equal to max context length in pretraining (e.g. θ=L_train) improves extrapolation but degrades interpolation (Section 6, F and G). However, our experiments show that enlarging θ only at inference time recovers interpolation performance while still enhancing extrapolation (Section G).
>
> This predictive and mechanistic understanding, as well as the FMRoPE method derived from it, is not present in Barbero et al.’s work.
> We believe this discovery will prompt reconsideration of the prevailing trend of setting θ to high values.
> I acknowledge that Section 3 shares many similarities, but everything thereafter, including the appendix, represents new findings revealed in this paper.
>
> > **Q1: Can you explain why Phi-3s performance drops so quickly. In particular, what about block sparse matrix makes this phenomenon go away.**
>
> Thank you for your question. Please see reply for W1.
>
> > **Q2: What is the key innovation with respect to the work of Barbero et al?**
>
> Thank you for your question. Please see reply for W2.
>
> ---
>
> We appreciate your detailed comments.
> If there are specific points in our responses that remain unclear or insufficient, we would be glad to provide further clarification.

---

### Public Comment · ~Yui_Oka1 · 2026-03-02

Dear Program Committee, Area Chair, and Reviewers,


In response to the comments from the reviewers and the Area Chair, we have revised the title to improve its clarity. We have also carefully addressed the issues raised during the review process in the revised manuscript.
For detailed explanations of the changes, please refer to our rebuttal comments.


We sincerely appreciate your time and constructive feedback.

---

### Meta-Review · Area_Chair_kq6r · 2026-01-07

**Summary:**

Reviewers acknowledged the paper's contributions: an empirical and theoretical analysis of RoPE frequency bands and their relationship to θ and training length, a closed-form predictor for optimal frequencies, and FMRoPE's improved extrapolation through θ-matching. However, they raised several concerns. The novelty appeared limited as the work primarily extends Barbero et al., experiments were conducted at small scale, and the initial evaluation focused heavily on perplexity without testing long-context reasoning capabilities (e.g., NIAH, RULER). Additional issues included unvalidated claims about weight decay effects and unexplained model anomalies such as those observed in Phi-3.

In response, the authors added NIAH results and downstream task evaluations, conducted 1B-scale experiments, analyzed layer and head stability, provided full-matrix analysis, and clarified their distinctions from prior work. While these additions strengthened the submission, concerns about large-scale applicability and generalizability remained inadequately addressed, leading to mixed reviews that leaned toward marginal acceptance.

**Reviewer Concerns:**

The authors addressed several concerns: they explained the Phi-3 anomaly with block-sparse attention details, clarified distinctions from Barbero et al. by highlighting new findings on when, where, and why frequency bands emerge along with their trade-offs, supplemented perplexity metrics with NIAH and downstream task results, provided practical heuristics for θ selection, removed the unvalidated weight decay claim, corrected equations and typos, extended their predictor to handle different θ values, and verified band stability across training epochs.

However, several issues remain unresolved: full validation at large scale (7B+ parameters) is still missing, pretraining effects need further investigation, perturbation analysis lacks depth (no refactoring tests), mismatches in θ across different training stages are unexplored, impacts on attention heads and cross-token dynamics require examination, and while the authors noted a minor change regarding variance proxy versus full eigenvalue analysis, a comprehensive investigation is still needed.

**Reviewer Scores:**

Reviewer aPpJ: Would maintain 8 (rebuttals clarify innovations vs Phi-3; strong practical insights).
Reviewer jmkZ: Would maintain 6 (new evals address scope but trade-off under-explored).
Reviewer 26SL: Would raise to 6 (FMRoPE applicability clearer but small-scale persists).
Reviewer K7WU: Would maintain 4 (bias in citation suspicion; novelties incremental).
Reviewer uqNo: Would maintain 4 (FMRoPE trade-off not superior).

---

### Decision · Program_Chairs · 2026-01-26

Accept (Poster)